# ARROW: Augmented Replay for RObust World models

**Abdulaziz Alyahya**                                    *abalyahya@imamu.edu.sa*
*Department of Information Systems*
*College of Computer and Information Sciences*
*Imam Mohammad Ibn Saud Islamic University (IMSIU)*

**Abdallah Al Siyabi**                                   *abdallah.alsyiabi@monash.edu*
*Department of Data Science & AI*
*Faculty of Information Technology*
*Monash University*

**Markus R. Ernst**                                      *m.ernst@unsw.edu.au*
*School of Computer Science and Engineering*
*Faculty of Engineering*
*University of New South Wales, Sydney*

**Luke Yang**                                            *luke.yang@monash.edu*
*Department of Data Science & AI*
*Faculty of Information Technology*
*Monash University*

**Levin Kuhlmann**                                       *levin.kuhlmann@monash.edu*
*Department of Data Science & AI*
*Faculty of Information Technology*
*Monash University*

**Gideon Kowadlo**                                       *gideon@cerenaut.ai*
*Cerenaut – https://cerenaut.ai*

**Reviewed on OpenReview:** *https://openreview.net/forum?id=3FK2tFwNwK*

## Abstract

Continual reinforcement learning challenges agents to acquire new skills while retaining previously learned ones with the goal of improving performance in both past and future tasks. Most existing approaches rely on model-free methods with replay buffers to mitigate catastrophic forgetting; however, these solutions often face significant scalability challenges due to large memory demands. Drawing inspiration from neuroscience, where the brain replays experiences to a predictive World Model rather than directly to the policy, we present *ARROW* (Augmented Replay for RObust World models), a model-based continual RL algorithm that extends DreamerV3 with a memory-efficient, distribution-matching replay buffer. Unlike standard fixed-size FIFO buffers, ARROW maintains two complementary buffers: a short-term buffer for recent experiences and a long-term buffer that preserves task diversity through intelligent sampling. We evaluate ARROW on two challenging continual RL settings: Tasks without shared structure (Atari), and tasks with shared structure, where knowledge transfer is possible (Procgen CoinRun variants). Compared to model-free and model-based baselines with replay buffers of the same-size, ARROW demonstrates substantially less forgetting on tasks without shared structure, while maintaining comparable forward transfer. Our findings highlight the potential of model-based RL and bio-inspired approaches for continual reinforcement learning, warranting further research. Code is available at https://github.com/Cerenaut/ARROW.

# 1 Introduction

The ability to continually acquire new skills while retaining old ones is central to intelligence. However, many AI systems suffer from catastrophic forgetting, where learning new tasks abruptly degrades earlier capabilities (McCloskey & Cohen, 1989; French, 1999). In order to deploy reinforcement learning (RL) agents in open-ended, sequentially changing environments, overcoming this limitation becomes essential.

## 1.1 The challenge of continual reinforcement learning

Classical RL optimizes expected return in a single, stationary environment (Sutton & Barto, 1998), enabling major successes in Atari, Go, and beyond (Mnih et al., 2015; Silver et al., 2017; Vinyals et al., 2019; Levine et al., 2016). Many real-world settings, however, require adaptation across sequential tasks. In *continual reinforcement learning* (CRL), an agent encounters a curriculum $\mathcal{T} = (\tau_1, \tau_2, \ldots, \tau_T)$, often without task boundaries or identifiers (Khetarpal et al., 2022).

While continual learning (CL) has long been studied in supervised learning (Kirkpatrick et al. (2017); Lopez-Paz & Ranzato (2017)), supervised pipelines can often mitigate non-stationarity by reshuffling and replaying a fixed dataset (French, 1999). In RL, a stationary data distribution may be unattainable: data are streamed, tasks may not reliably repeat, and the environment itself can be non-stationary.

Following Parisi et al. (2019), CL requires balancing stability, plasticity, and transfer. Agents must retain prior performance (stability) while learning new tasks efficiently (plasticity), and ideally reuse earlier knowledge to accelerate future learning (forward transfer) or improve earlier tasks (backward transfer), especially when tasks share dynamics or visual structure. These goals are inherently in tension (Parisi et al., 2019), forming the *stability–plasticity dilemma* (Mermillod et al., 2013).

## 1.2 Related work and limitations

Continual learning methods are commonly grouped into parameter regularization, architectural modularity, and rehearsal/replay. Parameter regularization constrains parameter updates to preserve weights important for prior tasks, as in Elastic Weight Consolidation (EWC) (Kirkpatrick et al., 2017). Modularity dedicates task-specific components or routes computation through task-specific pathways, exemplified by PathNet (Fernando et al., 2017) and Progress & Compress (P&C) (Schwarz et al., 2018). Replay methods interleave stored experiences with new data to rehearse previous tasks (Rolnick et al., 2019; Riemer et al., 2019; Chaudhry et al., 2021) and are among the most effective approaches, but naive replay scales poorly because retaining complete experience histories demands large memory (OpenAI et al., 2019).

The state-of-the-art model-free CRL methods such as CLEAR (Rolnick et al., 2019) combine large replay buffers with V-trace off-policy correction and behavior cloning for stability, while replay-buffer augmentations inspired by neuroscience (e.g., selective replay based on surprise or reward) indicate that matching the global training distribution can mitigate catastrophic forgetting without storing all experiences (Isele & Cosgun, 2018). More recent work has applied replay strategies to model-based agents under non-stationarity. For CRL specifically, Kessler et al. (2023) introduce Continual-Dreamer and systematically compare selective experience-replay methods on a DreamerV2 backbone. In an adjacent setting, adapting to a localized reward change within a single task rather than a multi-task curriculum, Rahimi-Kalahroudi et al. (2023) propose a *local forgetting* (LoFo) replay buffer that evicts samples near newly-observed states rather than the oldest, enabling DreamerV2 and PlaNet to handle local environment changes.

Each family targets a different axis than ARROW and is complementary rather than a drop-in alternative. (i–ii) Parameter-regularization (EWC) and modular methods (PathNet, P&C) act on network parameters, not the replay distribution, and require task boundaries or identifiers. (iii) CLEAR (Rolnick et al., 2019), the only task-agnostic method on the same axis as ARROW, is empirically validated at buffer capacities $\approx$10–900 times our $2^{19}$ observation budget; running it here would extrapolate beyond its validated regime.[1] Its behavior-cloning of the policy on replayed states is complementary to ARROW (Sec. 6). (iv) Prioritized experience replay (PER) is a sampling rule that slots into the FIFO half of our buffer, separating *what to keep*

---

[1]The original paper attributes degradation at its smallest tested size to over-fitting on limited stored examples.

(our contribution) from *how to sample it*. We therefore compare against *DreamerV3* (matched architecture, isolating the buffer's contribution) and *TES-SAC* (the strongest model-free agent at our memory budget; Sec. 4.2) at the same total budget; layering ARROW on top of (i)–(iv) is a natural follow-up.

### 1.3 The neuroscience-inspired alternative

Complementary Learning Systems (CLS) theory (Hassabis et al., 2017; Khetarpal et al., 2022) posits two interacting memory systems: a fast system that captures recent episodes and a slow system that builds structured knowledge. In this view, the hippocampus replays recent experiences to the neocortex, a slow statistical learner, reducing forgetting; the neocortex can be interpreted as forming a predictive World Model (Mathis, 2023). Yet in RL, replay is typically used to improve model-free policies rather than to train a World Model directly.

World models are central to model-based RL, predicting action consequences (Ha & Schmidhuber, 2018; Hafner et al., 2019) and underpin DreamerV1-V3 (Hafner et al., 2020; 2021; 2025). They have also been applied to continual RL (Nagabandi et al., 2019; Huang et al., 2021; Kessler et al., 2023). Notably, Kessler et al. (2022) showed that DreamerV2 (Hafner et al., 2021) with a persistent FIFO buffer can reduce forgetting. However, World Model CL approaches often rely on replay buffers with millions of high-dimensional samples, creating substantial memory and scalability constraints.

World models are a natural fit for replay because they support off-policy learning, consistent with neuroscientific motivation, but remain underexplored in memory-efficient continual settings. Thus, a key open question is: Can strategic, memory-efficient replay to World Models enable robust continual learning while retaining the sample efficiency of existing approaches?

### 1.4 Our contribution

Building on Yang et al. (2024), we introduce *ARROW* (Augmented Replay for RObust World models), a model-based continual RL algorithm that extends DreamerV3 (Hafner et al., 2025) with a dual-buffer replay mechanism (Isele & Cosgun, 2018): a short-term FIFO store paired with a long-term, distribution-matching store, operating under a fixed memory budget.

Our contribution is the *working recipe*, not a new sampling primitive. We show that operationalizing the CLS-theory split, a fast hippocampal short-term store paired with a slow, statistically-matched long-term store, inside a state-of-the-art world-model agent is sufficient to turn a strongly recency-biased system into a competent continual learner under a fixed memory budget. The empirical evidence is that the recipe as a whole, rather than any single component, produces the gains.

We evaluate ARROW in two continual learning regimes: (i) tasks *without shared structure*, where each task is a distinct environment and reward function, and (ii) tasks *with shared structure*, where common dynamics or visual features enable transfer (Khetarpal et al., 2022; Riemer et al., 2019)—reflecting practical settings such as a household robot acquiring related skills. Accordingly, we measure not only forgetting but also forward and backward transfer, and target scalability in memory and computation (Khetarpal et al., 2022; Chen & Liu, 2018). We benchmark against matched-memory model-based (DreamerV3) and model-free (TES-SAC) baselines.

ARROW substantially reduces forgetting on tasks without shared structure while maintaining comparable forward transfer, demonstrating that bio-inspired replay strategies can deliver robust continual learning with modest memory and advancing lifelong agents that continuously acquire and refine skills in open-ended environments. Source code, training scripts, task-suite configurations, and hyperparameters are available at https://anonymous.4open.science/r/ARROW-B6F2/.

## 2 Background

We consider continual learning (CL) in finite, discrete-time, partially observable reinforcement learning (RL) environments modeled as Partially Observable Markov Decision Processes (POMDPs) (Kaelbling

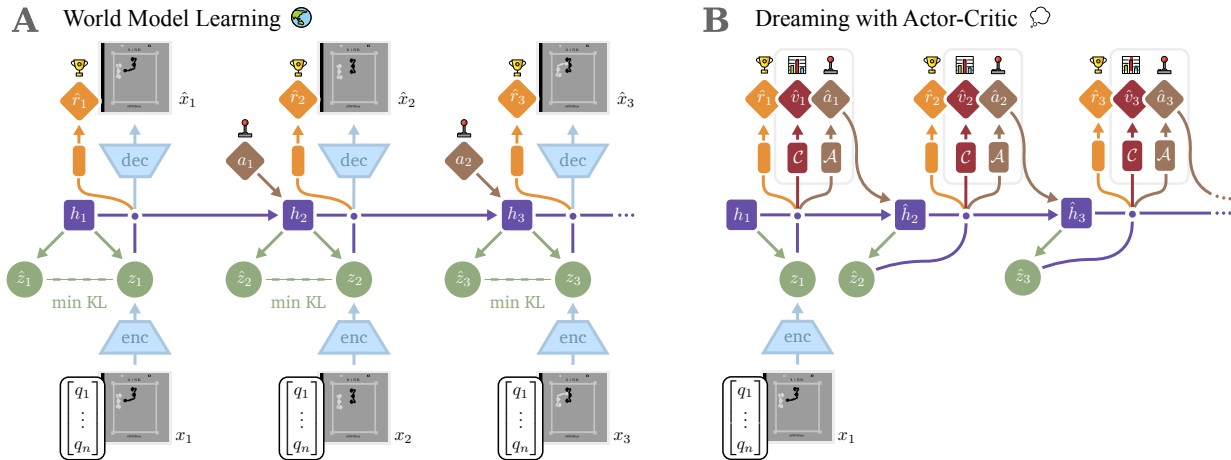

Figure 1: World Model Learning. (A) Images drawn from the replay buffer are encoded to and reconstructed from a latent space using a recurrent state space model. (B) Learning the policy is achieved with Actor ($\mathcal{A}$) and Critic ($\mathcal{C}$) networks applied to latent states "dreamt-up" by the model.

et al., 1998), a generalization of fully observable MDPs (Puterman, 1990). A POMDP is defined by $M = (\mathcal{S}, \mathcal{A}, p, r, \Omega, \mathcal{O}, \gamma)$, where $s_t \in \mathcal{S}$ evolves via $s_{t+1} \sim p(s_t, a_t)$ under actions $a_t \in \mathcal{A}$, yielding reward $r(s_t, a_t, s_{t+1}) \in \mathbb{R}$. The agent observes $\omega_t \in \Omega$ generated by $\omega_t \sim \mathcal{O}(s_t)$, and optimizes discounted returns with $\gamma \in (0, 1)$.

Actions are sampled from a stochastic policy $a_t \sim \pi(\omega_t)$, typically parameterized by a neural network $\pi_\theta$. Under a finite horizon $T$, the return is $R_t = \sum_{i=t}^{T} \gamma^{i-t} r(s_i, a_i, s_{i+1})$, and the objective is to maximize $\mathbb{E}_\pi[R_0 \mid s_0]$. RL methods may be model-free or model-based: model-free learning directly optimizes $\pi_\theta$, while model-based approaches learn a simulator (a *World Model*) of $p$ and $r$ from past rollouts and can use it to simulate trajectories for planning and action selection, or to train the policy from imagined rollouts. RL algorithms are also on- or off-policy; on-policy methods require fresh data from the latest $\pi_\theta$, whereas off-policy methods learn from previously collected samples despite policy mismatch (Espeholt et al., 2018). Off-policy replay buffers can be used to train World Models and improve sample efficiency, often yielding substantially better data efficiency than purely model-free optimization.

## 3 Augmented Replay for RObust World models (ARROW)

ARROW extends DreamerV3, which achieved state-of-the-art performance on several single-GPU RL benchmarks, not including explicit testing on continual RL. It comprises three components: a *World Model* of the environment, an *actor-critic controller* for decision making, and an *augmented replay buffer* that stores experience. The replay buffer is used to train the World Model, which then generates imagined ("dreamed") trajectories used to train the controller, enabling off-policy learning and data augmentation—useful in continual learning when environment interaction is limited. ARROW does not require explicit task identifiers, allowing for more flexible adaptation to changing environments.

Sections 3.1 (World model) and 3.2 (Actor-critic controller) recapitulate the DreamerV3 design and are included for self-containedness; they are not part of our contribution. The contribution begins at Sec. 3.3 (the augmented replay buffer), with the task-agnostic exploration and reward scaling treatments in Sec. 3.4 as supporting design choices.

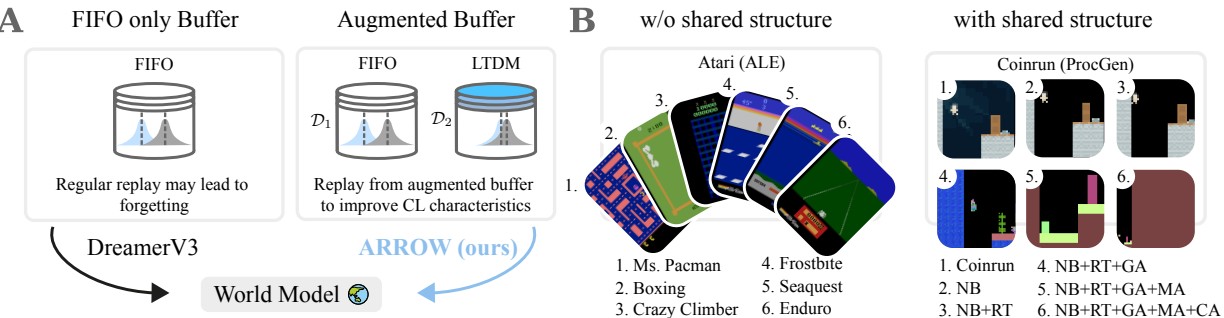

Figure 2: Experiment setup. (A) Augmented buffer used in ARROW. (B) Continual learning tasks with and without shared structure. NB: no background, RT: restricted themes, GA: generated assets, MA: monochrome assets, CA: centered agent.

## 3.1 World model

As with DreamerV3, ARROW uses a Recurrent State-Space Model (RSSM) (Hafner et al., 2019) to predict dynamics, see Fig. 1A. It maintains a deterministic hidden state $h_t$ and a stochastic latent state $z_t$ conditioned on observations $x_{1:t}$ and actions $a_{1:t}$. A GRU models the dynamics by predicting the deterministic state $h_{t+1} = \text{GRU}(h_t, z_t, a_t)$ and the stochastic state $\hat{z}_{t+1} = f(h_{t+1})$. The latent $z_t$ is inferred via a variational encoder $z_t \sim q_\theta(z_t \mid h_t, x_t)$, while the prior $\hat{z}_t \sim p_\theta(\hat{z}_t \mid h_t)$ is used for open-loop dreaming when posteriors are unavailable. We use a standard GRU with tanh activation, and set $z_t$ to 32 discrete units with 32 categorical classes.

The World Model state at time $t$ concatenates $h_t$ and $z_t$, yielding a Markovian representation. Training reconstructs images and rewards, using KL balancing (Hafner et al., 2021) to stabilize transitions.

## 3.2 Actor-critic controller

The actor and critic are MLPs that map World Model states to actions and value estimates, see Fig. 1B. They are trained entirely on imagined trajectories generated by the World Model ("dreaming"), using on-policy REINFORCE (Williams, 1992). Imagined trajectories are inexpensive and avoid additional environment interaction.

## 3.3 Augmented replay buffer

The augmented replay buffer, Fig. 2A, consists of a short-term FIFO buffer $\mathcal{D}_1$ (as in DreamerV3) and a long-term global distribution matching buffer $\mathcal{D}_2$, used in parallel and sampled uniformly for each minibatch. Each buffer stores $2^{18} \approx 262{,}000$ observations, for a total of $2^{19}$, half the $2^{20}$ used in the original DreamerV3 setup, with no noticeable performance loss on our benchmarks. All methods are matched at this $2^{19}$ budget in our experiments (Sec. 4.3). We use *memory-efficient* in this like-for-like sense: differences in continual-learning behavior therefore reflect how the budget is *allocated*, not its size. We ablate the FIFO/LTDM ratio (50/50 in the main experiments) at fixed total budget in Appendix A.1. To further reduce storage pressure, we store *spliced rollouts* instead of entire episodes.

**Short-term FIFO buffer** The FIFO buffer stores the most recent $2^{18}$ samples, ensuring the World Model trains on all incoming experience with a recency bias that improves convergence on the current task.

**Long-term global distribution matching (LTDM) buffer** Matching the global training distribution under limited capacity can reduce catastrophic forgetting (Isele & Cosgun, 2018). Our LTDM buffer also has capacity $2^{18}$, and stores a uniform random subset of 512 spliced rollouts. We use reservoir sampling by

assigning each rollout chunk a random key and maintaining a size-limited priority queue that retains the highest keys.

**Spliced rollouts** With small buffers, storing full episodes can yield too few unique trajectories, biasing training data and reducing World Model accuracy, especially in $\mathcal{D}_2$, which should retain coverage across tasks. We therefore splice rollouts into chunks of length 512, ensuring a controlled sampling granularity. The remaining rollouts with fewer than 512 states are concatenated with subsequent episodes, using a reset flag to mark boundaries. For efficiency, episodes are also truncated after a fixed number of steps so each iteration collects an identical number of environment steps. In practice, splicing provided a simple way to control granularity without harming performance.

### 3.4 Task-agnostic exploration and reward scaling

In tasks without shared structure, environments can differ sharply in dynamics, visuals, and reward scales, making exploration difficult without task IDs; policies trained on earlier tasks may be insufficiently stochastic on new tasks. We apply fixed-entropy regularization (as in DreamerV3) during actor-critic training. This mitigates exploration issues without adding an explicit exploration system such as Plan2Explore (Sekar et al., 2020).

**Reward scaling without task IDs.** Training a single actor across environments with rewards of vastly different magnitudes produces correspondingly mismatched advantage magnitudes: gradient updates become dominated by the highest-return tasks and the policy effectively ignores the others. Standard remedies include PopArt-style return normalization (van Hasselt et al., 2016; Hessel et al., 2019) and DreamerV3's symlog reward standardization (Hafner et al., 2025). Per-task reward magnitudes in our Atari suite span three orders of magnitude (0.001 on Crazy Climber to 1 on Boxing; Tab. A.15), and in development, learning on lower-magnitude tasks suffered without per-task scaling. We therefore use a static, linear, per-task reward scale derived from single-task baselines (Tab. A.15): (a) it keeps the comparison interpretable, (b) the scales are fixed offline and so do not require task IDs at training time, and (c) automatic non-linear advantage rescaling that we tried hurt single-task performance. This issue is specific to suites without shared structure; our CoinRun variants share a reward structure and require no scaling. See Sec. 6 for the trade-offs.

## 4 Experiments

### 4.1 Environments

**Atari (without shared structure)** For environments without shared structure, we selected six diverse Atari games from the ALE's v5 configurations spanning different visual modalities and gameplay dynamics (Bellemare et al., 2013). We chose Ms. Pac-Man, Boxing, Crazy Climber, Frostbite, Seaquest, and Enduro, see Fig. 2B. We presented the tasks to the agent in this specific but randomly chosen order (henceforth referred to as *default task order*) and followed (Machado et al., 2018) in using sticky actions.

**CoinRun (shared structure)** To construct a suite of continual learning tasks with shared structure, we used the CoinRun environment from OpenAI Procgen as the base (Cobbe et al., 2020). We then introduced six progressive visual and behavioral perturbations, as shown in Fig. 2B and described in Appendix Tab. A.1.

### 4.2 Training configurations and baselines

To evaluate continual learning behavior across both Atari and CoinRun, we adopted three training configurations. Training and evaluation details are summarized in Appendix Tab. A.2.

**Default task order:** The agent is trained on all tasks once, following the default order shown in Fig. 2B for each of the two benchmarks.

**Reversed task order:** The agent is trained on all tasks once but in *reverse order*. For Atari: Enduro $\rightarrow$ Seaquest $\rightarrow$ Frostbite $\rightarrow$ Crazy Climber $\rightarrow$ Boxing $\rightarrow$ Ms. Pac-Man. For CoinRun: CA $\rightarrow$ MA $\rightarrow$ GA $\rightarrow$ RT $\rightarrow$ NB $\rightarrow$ CoinRun.

**Two-cycle training:** This configuration uses the default task order but splits the total training budget into *two cycles*, with identical total environment steps to the two previous configurations. This allows us to examine *relearning*, *retention*, *cross-cycle adaptation*, and quantify performance recovery when a task is revisited.

The default and reverse orderings are chosen to *bracket* the space of single-pass curricula: as opposite orderings they probe the most dissimilar single-pair conditions. Task ordering can produce qualitatively different forgetting profiles (Rahimi-Kalahroudi et al., 2023, Appendix G), and our baselines show order-induced shifts (Sec. 5.1); the qualitative robustness of our findings across this pair is itself evidence that the conclusions are not artefacts of one curriculum. A larger sweep of random orderings would be more rigorous still and is left to future work alongside the per-task-budget and natural-curriculum sweeps in Sec. 6.

To measure the efficacy of the augmented replay buffer, we compared ARROW to DreamerV3 and model-free SAC, each with equal memory allowance for their respective replay buffers. We chose SAC as the strongest model-free continual-RL candidate at our memory budget. The closest model-free CRL alternative, CLEAR (Rolnick et al., 2019), is validated at replay capacities of 5M–450M frames against $\sim$900M training frames, with the authors attributing the performance drop at its smallest setting to over-fitting on limited stored examples. ARROW's total replay budget is $2^{19} \approx 0.5$M observations, roughly an order of magnitude below CLEAR's smallest validated regime; using it as a baseline here would test it in an extrapolated regime rather than constitute a fair comparison. For SAC, we adopted the Target Entropy Scheduled SAC (TES-SAC) variant, an extension proposed by Xu et al. (2021) that dynamically schedules the entropy target for improved stability and exploration. TES-SAC suits long, multi-task curricula: its scheduled target entropy gives higher exploration early and lower-entropy exploitation later, useful when transitioning between visually and dynamically distinct tasks. As discussed in Sec. 5.1, TES-SAC fails to reach baseline competency on Atari at this memory budget; we retain it because the resulting asymmetry (low forgetting paired with low absolute return) is itself diagnostic of how strict the same-budget regime is for model-free agents, and of why forgetting must be read alongside ACC and WC-ACC. We also ran single-task baselines that were used for evaluation metrics and normalization.

### 4.3 Replay memory allowance

All methods are compared under an equal replay memory budget. Replay buffers store *sequences* (spliced rollouts) of length $T = 512$; a capacity of 512 sequences corresponds to $512 \times 512 = 262{,}144 = 2^{18}$ observations.

For ARROW, we combine a short-term FIFO buffer and a long-term global distribution-matching (LTDM) buffer, each with capacity 512 sequences of length $T = 512$. Consequently,

$$N_{\text{ARROW}} = 2 \times 512 = 1024$$

and

$$T \times N_{\text{ARROW}} = 512 \times 1024 = 524{,}288 = 2^{19}.$$

DreamerV3 and TES-SAC use a single FIFO replay buffer with capacity 1024 sequences of length $T = 512$:

$$N_{\text{DV3/TES-SAC}} = 1024 \quad \Rightarrow \quad T \times N_{\text{DV3/TES-SAC}} = 512 \times 1024 = 524{,}288 = 2^{19}.$$

### 4.4 Evaluation metrics

Performance in continual RL is multidimensional, so we evaluate it along three complementary axes. Per-task return curves ground the comparison in absolute task performance. End-of-training outcomes across all tasks, ACC, min-ACC, and WC-ACC (Lange et al., 2023) capture the deployment-relevant view. *Forgetting*

and *forward transfer* (Kessler et al., 2023) characterize the dynamics that produced both. For our two-cycle setting, we further report *Recovery* and introduce a new cross-cycle metric, *maximum forgetting* (Max-F).

We evaluated performance by normalizing episodic reward to two single-task baselines: ARROW and a random agent.

### 4.4.1 Normalized rewards

We define an ordered suite of tasks $\mathcal{T} = (\tau_1, \tau_2, \ldots, \tau_T)$, where $\tau_i$ denotes the $i$-th task in the curriculum and $i \in \{1, \ldots, T\}$. Performance in task $\tau \in \mathcal{T}$ after $n$ steps in single-task experiments is given by $p_{\mathrm{ST}_\tau}(n)$ and in CL by $p_\tau(n)$. Agents were trained on each task for $n = N$ environment steps in single-task and CL experiments. For each task $\tau \in \mathcal{T}$, we calculated the normalized reward using $p_{\mathrm{ST}_\tau}(0)$ and $p_{\mathrm{ST}_\tau}(n)$:

$$q_\tau(n) = \frac{p_\tau(n) - p_{\mathrm{ST}_\tau}(0)}{p_{\mathrm{ST}_\tau}(n) - p_{\mathrm{ST}_\tau}(0)}. \tag{1}$$

A normalized score of 0 corresponds to random performance and a score of 1 corresponds to the performance when trained on only that task.

### 4.4.2 Forgetting (Backward transfer)

Average forgetting for each task is the difference between performance after training on a given task and performance at the end of all tasks. Average forgetting over all tasks is defined as:

$$F = \frac{1}{T} \sum_{i=1}^{T} \bigl( q_{\tau_i}(i \times N) - q_{\tau_i}(T \times N) \bigr). \tag{2}$$

A lower value indicates improved *stability* and a better continual learning method. A negative value implies that the agent has managed to gain performance on earlier tasks, thus exhibiting *backward transfer*.

### 4.4.3 Forward transfer

The forward transfer for a task is the normalized difference between performance in the CL and single-task experiments. The average over all tasks is defined as:

$$FT = \frac{1}{T} \sum_{i=1}^{T} \frac{S_{\tau_i} - S_{\mathrm{ST}_{\tau_i}}}{S_{\mathrm{ST}_{\tau_i}}}, \tag{3}$$

where

$$S_{\tau_i} = \frac{1}{N} \sum_{n=1}^{N} q_{\tau_i}((i-1) \times N + n), \tag{4}$$

$$S_{\mathrm{ST}_{\tau_i}} = \frac{1}{N} \sum_{n=1}^{N} q_{\mathrm{ST}_{\tau_i}}(n). \tag{5}$$

The larger the forward transfer, the better the continual learning method. A positive value implies effective use of learned knowledge from previous environments and, as a result, accelerated learning in the current environment. When each task is not related to the others, no positive forward transfer is expected. In this case, forward transfer of 0 represents optimal *plasticity*, and negative values indicate a barrier to learning newer tasks from previous tasks.

### 4.4.4 Stability-plasticity metrics

To characterize the stability-plasticity trade-off in continual learning, we follow Lange et al. (2023) and analyze three complementary metrics: Average accuracy (ACC), average minimum accuracy (min-ACC) and Worst-case Accuracy (WC-ACC). These assess how well the agent learns the current task (*plasticity*) while retaining knowledge of previous tasks (*stability*).

**Average accuracy (ACC)**   After completion of task $\tau_k$, ACC measures performance on all tasks encountered up to that point:

$$\text{ACC}_{\tau_k} = \frac{1}{k} \sum_{i=1}^{k} q_{\tau_i}(t_k). \tag{6}$$

Here, $q_{\tau_i}(t_k)$ is the normalized performance on task $\tau_i$ evaluated at the end of task $\tau_k$.

**Average minimum accuracy (min-ACC)**   min-ACC tracks the worst performance each previously learned task attains after it has been learned (average of each task's own minimum):

$$\text{min-ACC}_{\tau_k} = \frac{1}{k-1} \sum_{i=1}^{k-1} \min_{t_i < n \leq t_k} q_{\tau_i}(n). \tag{7}$$

For each earlier task $\tau_i$, we take the minimum normalized performance over evaluation steps $n$ after $t_i$ and up to $t_k$, then average these minima over the $k-1$ previous tasks.

**Worst-case accuracy (WC-ACC)**   WC-ACC can be evaluated at every training iteration. At iteration $n$ within task $\tau_k$:

$$\text{WC-ACC}_n = \frac{1}{k} q_{\tau_k}(n) + \left(1 - \frac{1}{k}\right) \text{min-ACC}_{\tau_k}. \tag{8}$$

The first term, $\frac{1}{k} q_{\tau_k}(n)$, reflects performance on the current task and therefore measures plasticity. The second term, weighted by $1 - \frac{1}{k}$, incorporates the minimum performance that each previously learned task ever achieved through min-ACC, which accounts for the minimum accuracy that each earlier task ever reached. Unlike ACC, which is only defined at task boundaries, WC-ACC provides a continuous view of how the agent trades off learning the present task while maintaining stability on earlier ones throughout training.

### 4.4.5   Sample efficiency

Another important factor for practical applications, especially where agents operate in the real world (as opposed to simulation), is sample efficiency. Therefore, we analyzed how quickly each method reaches performance thresholds. Sample efficiency is measured as the median number of environment frames required to reach 85% of the maximum median performance across all methods. Any threshold-of-max convention favors higher-ceiling methods; we use a shared cross-method maximum as the fairest common benchmark. For context, the per-method maxima are in Tab. A.7 and Tab. A.8.

Formally, let $P_t^{(m)}$ denote the median normalized performance of method $m$ at frame $t$, and let $P^* = \max_{m,t} P_t^{(m)}$ be the maximum performance achieved by any method during training. The sample efficiency for method $m$ is defined as:

$$\text{SE}_m = \min \left\{ t : P_t^{(m)} \geq 0.85 \cdot P^* \right\}. \tag{9}$$

### 4.4.6   Two-cycle metrics

We extended our evaluation to a two-cycle setting that uses the same overall training budget as the one-cycle curriculum. The total number of environment steps allocated to each task is divided into two equal exposures: each task $\tau_i$ is first encountered during cycle 1 and then revisited for the remaining budget during cycle 2.

**Maximum forgetting (Max-F)**   To quantify the worst degradation that occurs between the first and second exposures of a task, we compute the maximum forgetting experienced by each task $\tau_i$ across the two-cycle curriculum:

$$t_i^{(1)} = i \times N, \qquad t_i^{(2)} = (T \times N) + (i-1) \times N, \qquad t_i^{(3)} = (T \times N) + i \times N, \tag{10}$$

$$\text{Max-F}_{\tau_i} = q_{\tau_i}\big(t_i^{(1)}\big) - q_{\tau_i}\big((t_i^{(2)})^-\big), \tag{11}$$

where $q_{\tau_i}(t_i^{(1)})$ denotes the normalized performance at the end of the first exposure to task $\tau_i$ in cycle 1, and $q_{\tau_i}((t_i^{(2)})^-)$ denotes the last evaluation *immediately before* the second exposure of the same task during cycle 2. We write $x^-$ to denote the final evaluation step strictly before step $x$; in particular, $(t_i^{(2)})^-$ is the last evaluation of task $\tau_i$ immediately before its second exposure begins at step $t_i^{(2)}$. The difference measures how much performance on task $\tau_i$ has deteriorated while the agent is learning other tasks.

Using Max-F compares forgetting at an equivalent point for each task: after the agent has completed training on *all* other tasks in cycle 1 but *before* relearning begins in cycle 2. In this interval, the agent is continuously trained on tasks $\tau_{i+1}, \ldots, \tau_T$. As a result, Max-F captures how resilient a task's knowledge remains when the agent has been exposed to the maximum possible amount of interference from all other tasks in the curriculum.

**Recovery**  The recovery metric quantifies how much performance on task $\tau_i$ is regained after the agent relearns the task during cycle 2, where $t_i^{(3)}$ denotes the end of the second exposure to task $\tau_i$:

$$\mathrm{Rec}_{\tau_i} = \frac{q_{\tau_i}(t_i^{(3)})}{q_{\tau_i}(t_i^{(1)})}. \tag{12}$$

### 4.5  Ablation: ARROW buffer-ratio variants (AR25, AR50, AR75)

To probe which parts of the augmented-buffer design are critical, we held the total replay budget fixed at $2^{19}$ observations and trained ARROW on all environments and settings under three FIFO/LTDM splits: AR25 (**75/25**), AR50 (**50/50**) (the configuration used in the main experiments), and AR75 (**25/75**). The total memory budget, network architecture, training schedule, and all other hyperparameters were held fixed; only the FIFO/LTDM allocation varies between conditions.

This sweep spans the design axis from a near-FIFO buffer (AR25) to a near-LTDM buffer (AR75), with AR50 as the symmetric midpoint. Because the total budget is held fixed, any difference in continual-learning behavior between variants reflects how the budget is *allocated* rather than its size. Per-suite results are reported in Sec. 5.3 and broken down in Appendix Tab. A.5 and Tab. A.6.

## 5  Results

### 5.1  Tasks without shared structure: Atari

Median normalized performance is shown in Fig. 3, while other metrics are visualized in Fig. 4. Detailed numerical results (median [IQR]) are provided in Appendix Tab. A.3.

ARROW nearly eliminates catastrophic forgetting, regardless of task order, while DreamerV3 suffers severe forgetting whenever a new task is introduced. TES-SAC's low forgetting scores (Fig. 4A,B) are an artefact of low absolute performance, not retention: it largely fails to learn Atari at this budget (median raw returns are close to the random-policy baselines in Tab. A.15), so there is nothing to forget. This is itself diagnostic, underscoring how strict the same-budget regime is for model-free agents and why forgetting must be read alongside ACC and WC-ACC.

**Default task order.** ARROW reduces forgetting by nearly twenty-fold compared to DreamerV3 (Fig. 4A). DreamerV3 has a clear edge in forward transfer, reflecting that its unconstrained buffer allows faster initial learning on new tasks, whereas ARROW's augmented buffer introduces a plasticity cost. Crucially, ARROW achieves the best overall stability–plasticity trade-off, with a positive WC-ACC against negative values for both baselines.

**Reversed task order.** The default-order conclusions are not order-specific: under the reverse curriculum (Fig. 3B, Fig. 4B), ARROW's forgetting stays low while DreamerV3 again forgets severely, and ARROW retains the highest WC-ACC. The qualitative ranking is identical to the default order, so we present these results primarily as a robustness check.

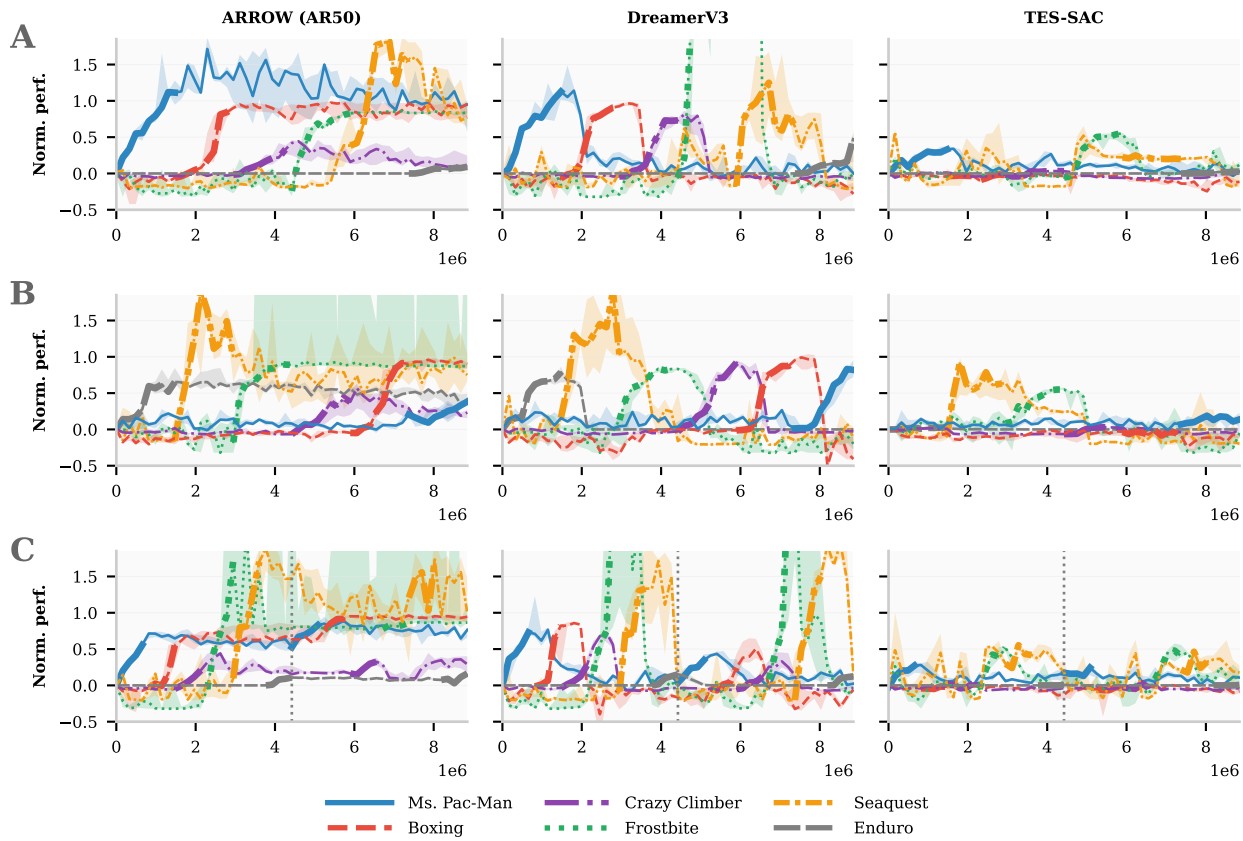

Figure 3: **Atari** median normalized performance (Eq. 1). Shaded area depicts 0.25 and 0.75 quartiles of 5 seeds. Bold line segments indicate training of task. (A) Default order of tasks (one-cycle). (B) Reversed order of tasks (one-cycle). (C) Default order of tasks (two-cycle). The dotted, vertical line marks the end of cycle 1 and the beginning of cycle 2.

**Two-cycle training.** The two-cycle setting reveals ARROW's most distinctive strength: recovering and even surpassing prior performance when tasks are revisited (Fig. 3C, Fig. 4C). ARROW's maximum forgetting (Max-F) is very small, an order of magnitude below DreamerV3, with TES-SAC sitting between the two. ARROW is also the only method to record a positive WC-ACC. Since two-cycle training most closely resembles real deployments (agents repeatedly cycling between a small set of tasks), we view it as the most practically informative of the three settings.

### 5.2 Tasks with shared structure: Procgen CoinRun

Median normalized performance is shown in Fig. 5, while other metrics are visualized in Fig. 6. Detailed numerical results (median [IQR]) are provided in Appendix Tab. A.4.

When tasks share visual and structural features, all methods forget less than in Atari, with ARROW in particular achieving near-zero forgetting in the reversed order and essentially zero maximum forgetting in the two-cycle setting. Because forgetting is generally lower across the board, the distinguishing factor becomes the stability–plasticity balance: ARROW attains the highest WC-ACC in every CoinRun configuration, pairing strong forward transfer with reliable retention of earlier tasks.

**Default task order.** Both model-based methods show strong forward transfer on CoinRun variants (Fig. 6A), with DreamerV3 ahead. ARROW's slower per-task rise (Fig. 5A) is the expected cost of mixing

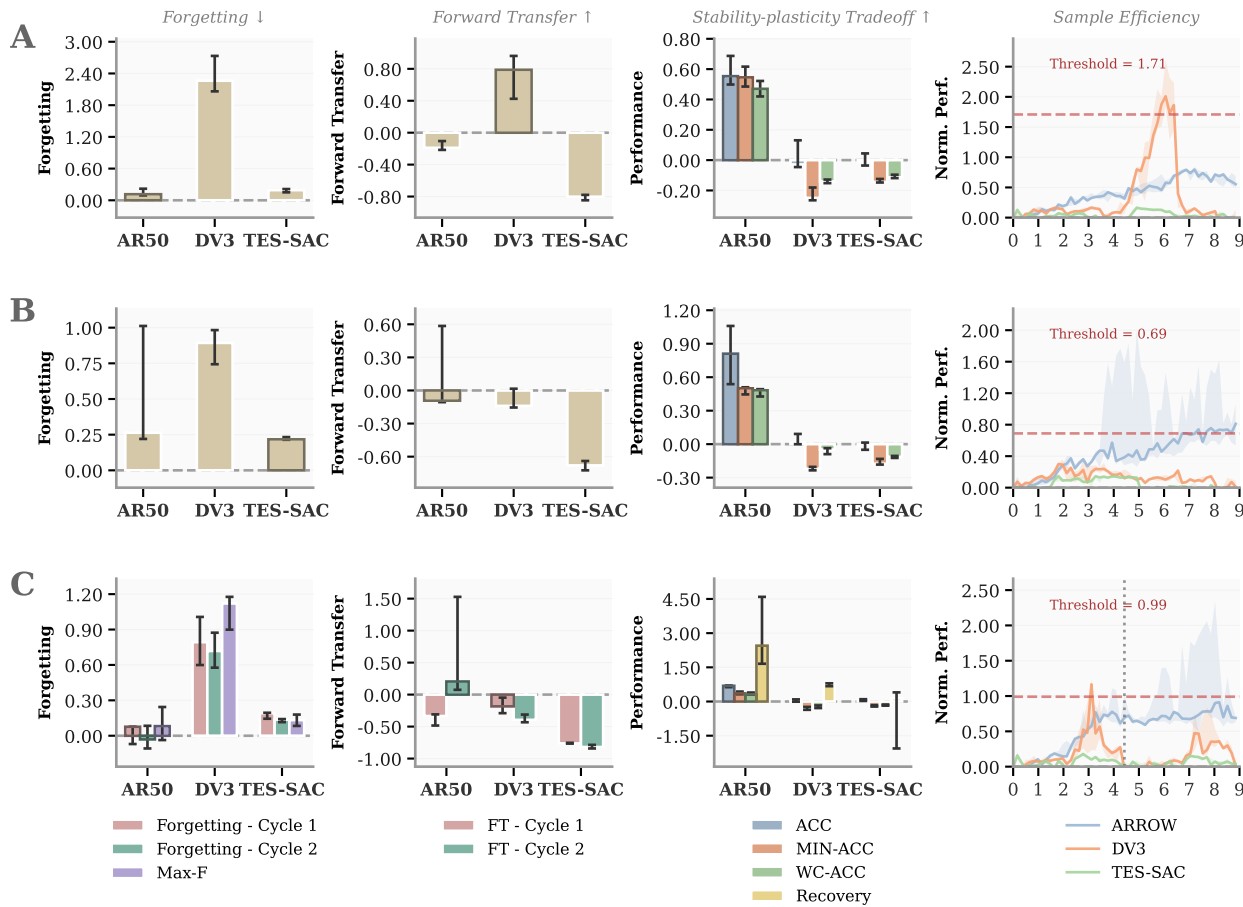

Figure 4: **Atari** metrics shown as median with (0.25 - 0.75) quartile confidence intervals, across 5 seeds, and calculated using normalized scores (Eq. 1). (A) Default task order (one-cycle). (B) Reversed task order (one-cycle). (C) Default task order (two-cycle).

prior-task transitions into every update, the same mechanism that yields the retention gains. ARROW achieves the better trade-off overall, with the highest WC-ACC. TES-SAC also benefits from the shared structure here, showing minimal forgetting and substantial forward transfer, though it still trails ARROW on WC-ACC.

**Reversed task order.** The reversed curriculum yields the same qualitative ordering (Fig. 5B, Fig. 6B): ARROW's forgetting drops to effectively zero, forward transfer rises (DreamerV3 still ahead), and ARROW achieves the strongest overall profile. As in Atari, we present this primarily as confirmation that the default-order conclusions are not order-specific.

**Two-cycle training.** In the two-cycle setting (Fig. 5C, Fig. 6C), ARROW's behavior is also distinctive: maximum forgetting is essentially zero, with the IQR straddling zero, indicating that per-task performance is largely retained between the first and second exposures. DreamerV3 forgets noticeably; TES-SAC is also near zero. All three methods achieve recovery above 1.0, confirming that revisiting CoinRun tasks is universally beneficial; ARROW stands out with by the strongest WC-ACC.

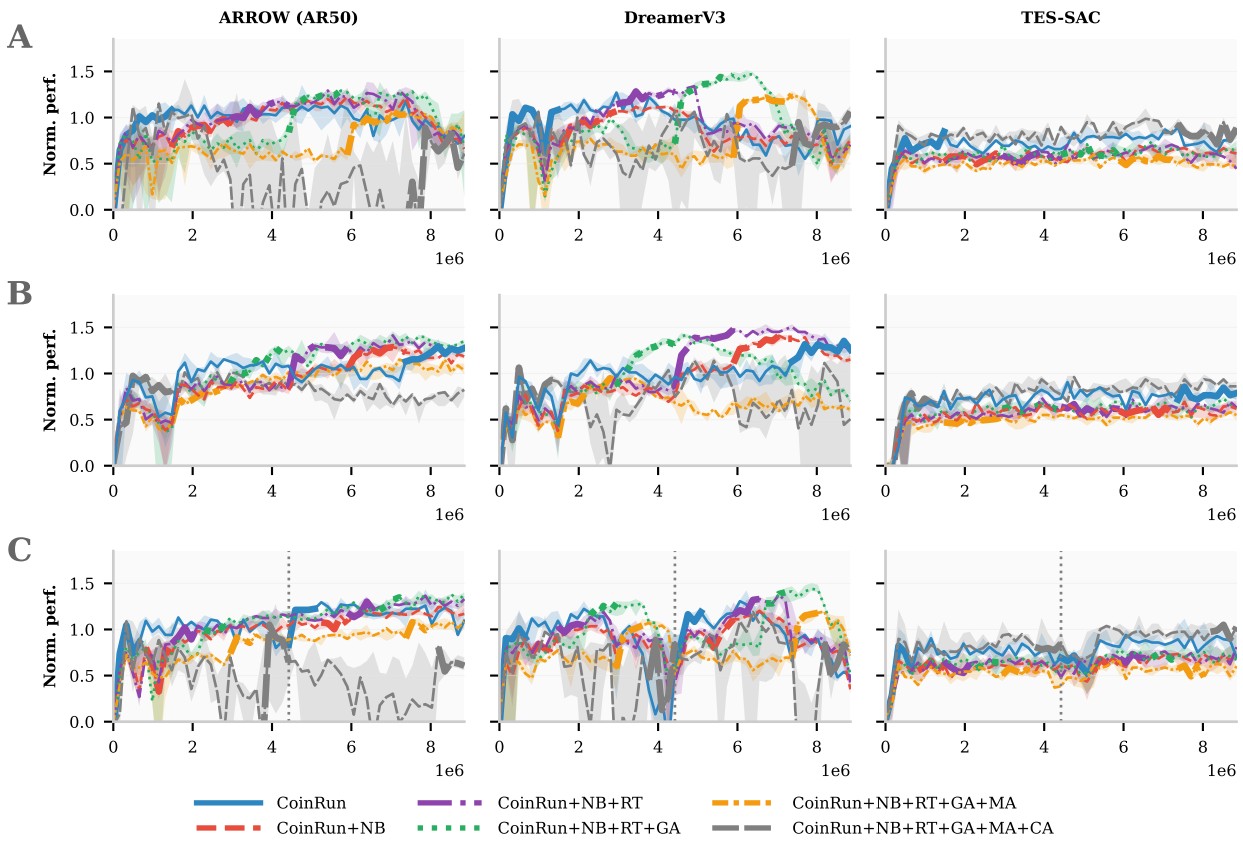

Figure 5: **CoinRun** median normalized performance (Eq. equation 1). Shaded area depicts 0.25 and 0.75 quartiles of 5 seeds. Bold line segments indicate training of task. (A) Default order of tasks (one-cycle). (B) Reversed order of tasks (one-cycle). (C) Default order of tasks (two-cycle). The dotted vertical line marks the end of cycle 1 and the beginning of cycle 2.

### 5.2.1 Continual learning sample efficiency

The last columns of Fig. 4 and Fig. 6 illustrate how quickly each method reaches a performance threshold of 85%. The non-shared tasks (Atari) show dramatic task-order sensitivity as the peak performances vary widely (2.01 for default, 0.81 for reversed and 1.17 for two-cycle).

In the default task order, only DreamerV3 reaches the threshold (4/5 seeds); neither ARROW nor TES-SAC reaches it in any seed. In the reversed task order, only ARROW reaches the threshold (3/5 seeds). For the two-cycle training both ARROW and DreamerV3 reach the 85% threshold, with DreamerV3 marginally more sample-efficient than ARROW. For detailed statistics including interquartile ranges, see Appendix Tab. A.7.

As for the tasks with shared structure, we observe greater consistency between the model-based approaches, where for each task configuration both ARROW and DreamerV3 consistently reach the threshold in all 5 seeds. TES-SAC fails to reach any threshold across all given CoinRun task configurations.

We observe that the sample efficiency for ARROW is sensitive to task order. In the default order ARROW and DreamerV3 look broadly similar: both fluctuate, peak around the same level, and drop, although DreamerV3 crosses the 85% threshold first. In the reversed and two-cycle orders ARROW is more consistent, reaches its plateau sooner, and rises higher overall. For detailed statistics including interquartile ranges, see Tab. A.8.

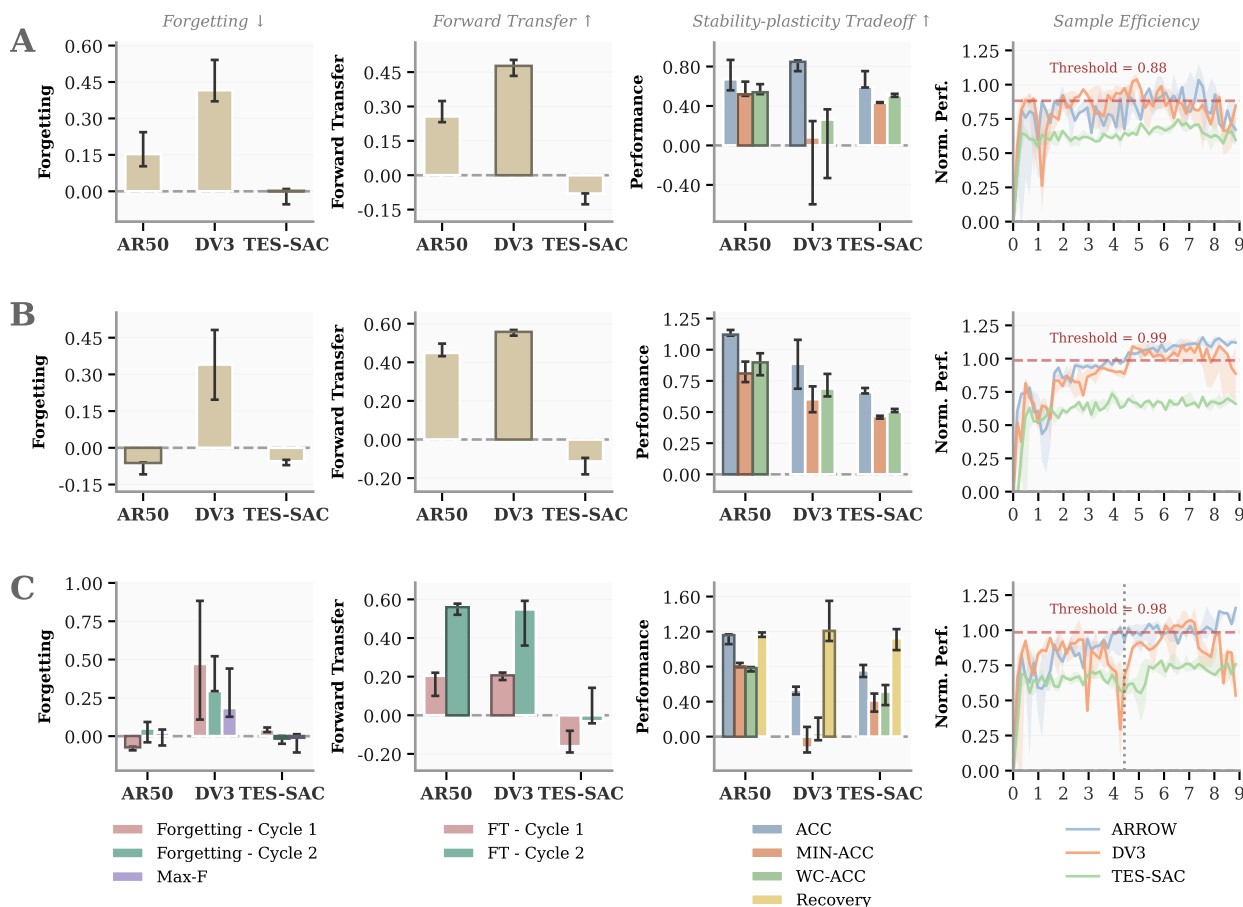

Figure 6: **CoinRun** metrics shown as median with (0.25 - 0.75) quartile confidence intervals, across 5 seeds, and calculated using normalized scores, Eq. 1. (A) Default order of tasks (one-cycle). (B) Reversed order of tasks (one-cycle). (C) Default order of tasks (two-cycle).

## 5.3  Buffer-ratio ablation (AR25, AR50, AR75)

Following the setup in Sec. 4.5, median normalized performance is shown in Fig. 7 and Fig. 8; per-suite metric bars are deferred to Appendix A.1 (Fig. A.1, Fig. A.2), and detailed numerical results (median [IQR]) are provided in Appendix Tab. A.5 and Tab. A.6.

**Tasks without shared structure: Atari.** The sweep traces a stability–plasticity axis: increasing LTDM share from AR25 to AR75 tends to reduce forgetting while shrinking forward transfer, with AR50 sitting between the two extremes. In the default task order, AR75 attains the lowest forgetting and the highest WC-ACC, while AR25 retains the most plasticity at the cost of the highest forgetting, and AR50 sits between the two. The reversed order yields a similar picture, with AR50 best on WC-ACC and AR75 best on forgetting. In the two-cycle setting, AR50 again leads on WC-ACC and AR75 minimizes maximum forgetting. Headline differences across the three variants are modest, however, and seed-to-seed IQRs overlap substantially.

**Tasks with shared structure: Procgen CoinRun.** When tasks share structure, forgetting stays low across all three variants and most pairwise gaps fall within the seed-to-seed IQRs. In the default task order, AR25 leads on every metric, AR50 has the highest forgetting and lowest WC-ACC, and AR75 sits between them. The reversed order compresses the ranking further—all three forget near zero, with AR50 best on forgetting and forward transfer and AR25 highest on WC-ACC (AR50 close behind). In the two-cycle setting,

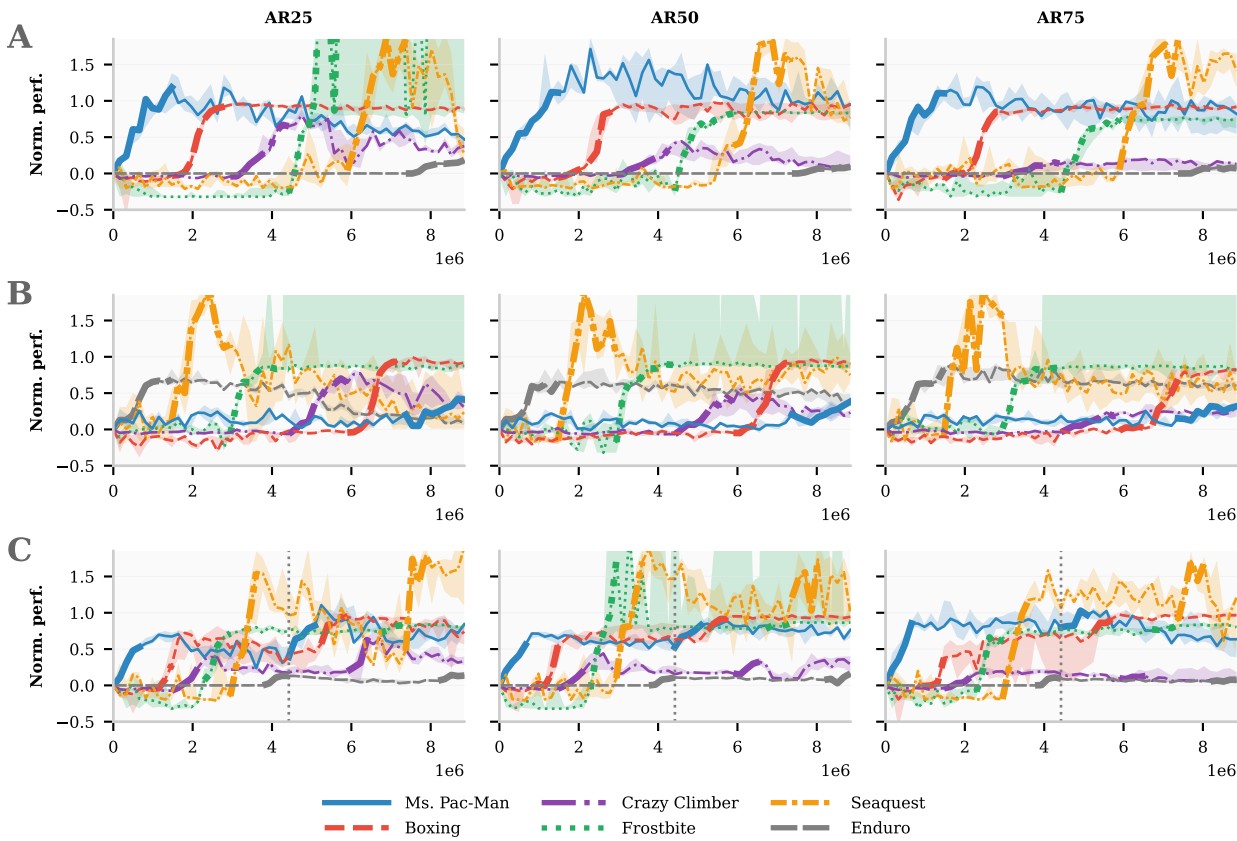

Figure 7: **Atari** median normalized performance (Eq. 1) of the ARROW buffer-ratio variants AR25, AR50, and AR75. Shaded area depicts 0.25 and 0.75 quartiles of 5 seeds. Bold line segments indicate training of task. (A) Default order of tasks (one-cycle). (B) Reversed order of tasks (one-cycle). (C) Default order of tasks (two-cycle). The dotted, vertical line marks the end of cycle 1 and the beginning of cycle 2.

AR50 yields the best WC-ACC and AR75 the lowest maximum forgetting; all three variants achieve recovery above 1, confirming that revisiting CoinRun tasks is universally beneficial regardless of the FIFO/LTDM allocation.

Taken together, differences across AR25, AR50, and AR75 are typically of the same order as the seed-to-seed variability, and no single variant dominates every task order within either suite. We adopt AR50 as the default for the main experiments because it remains competitive in every setting without being the strongest extreme on either axis; the broader takeaway is that the augmented-replay mechanism, rather than the specific FIFO/LTDM ratio, is the central design choice.

## 6 Discussion

The results demonstrate that ARROW tends to be more stable and trades off stability and plasticity better than the model-free and model-based baselines. Our results confirm that augmented replay to a World Model in a model-based approach provides a strong foundation for continual RL.

For tasks without shared structure, ARROW substantially reduces forgetting compared to DreamerV3. This suggests that the distribution-matching principle that ARROW is built upon is able to preserve World Model accuracy across previous and current tasks. Moreover, this is in stark contrast to the results obtained for DreamerV3 which exhibited catastrophic forgetting on every novel Atari task. TES-SAC also showed

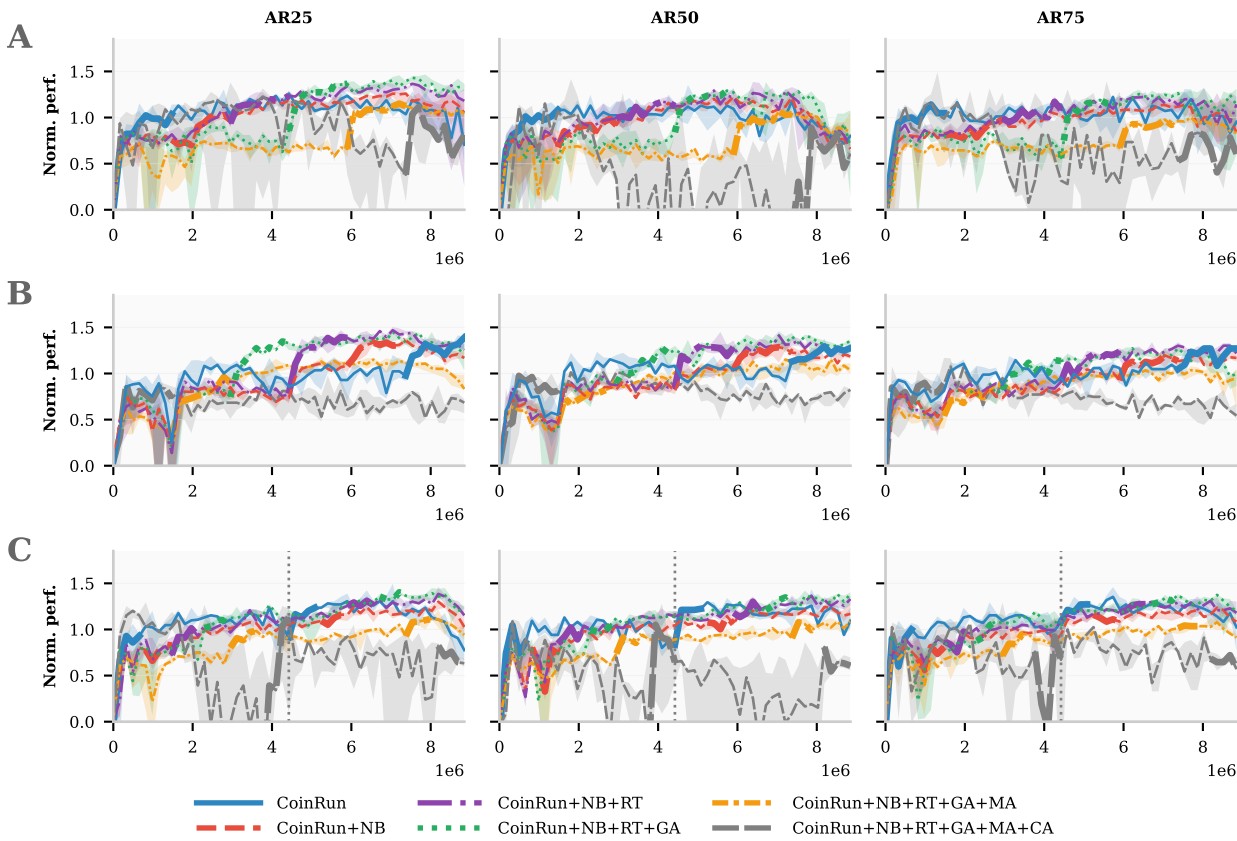

Figure 8: **CoinRun** median normalized performance (Eq. equation 1) of the ARROW buffer-ratio variants AR25, AR50, and AR75. Shaded area depicts 0.25 and 0.75 quartiles of 5 seeds. Bold line segments indicate training of task. (A) Default order of tasks (one-cycle). (B) Reversed order of tasks (one-cycle). (C) Default order of tasks (two-cycle). The dotted vertical line marks the end of cycle 1 and the beginning of cycle 2.

favorable forgetting statistics, but as noted in Sec. 5.1 this is an artifact of low absolute performance (raw returns close to the random-policy baselines in Tab. A.15) rather than retention, and must be read alongside ACC and WC-ACC. We ran a grid search over the main TES-SAC hyperparameters at the same memory budget but found no configuration that reliably learns Atari across seeds; we flag this as a limitation of model-free methods under our control rather than of TES-SAC itself. The same multi-dimensional reading applies to ARROW vs DreamerV3 on Atari. On across-tasks, end-of-training metrics, ARROW achieves substantially higher WC-ACC than either baseline, both of which record negative values. On raw per-task return the picture is different, as expected from retention-oriented methods: on Frostbite and Seaquest, DreamerV3 reaches higher peak per-task return during that task's training window (Fig. 3). ARROW trades slightly lower peak per-task return for substantially better retention across tasks—an appropriate trade-off for many deployment scenarios.

On tasks with shared structure, ARROW showed decreased forgetting. Notably, the magnitude of forgetting is tightly linked to the order of the presented tasks. In fact, the very last task of the default order "+CA" causes the camera to no longer be centered on the agent. This appears to be a significant departure from the other shared structure environments and causes a drop in normalized performance. When the order is reversed the near-zero forgetting of ARROW reappears. When we applied DreamerV3 to the shared structure tasks (CoinRun), we observed the variance was very high, resulting in low minimum average accuracy (min-ACC). ARROW mitigates that variance by making effective use of the augmented replay buffer, where past experiences stabilize the training.

In addition, ARROW was able to learn multiple tasks with highly varying magnitude of rewards, without task identifiers, which is an important and difficult challenge on its own.

The two-cycle setting resembles common continual-RL deployment scenarios, where an agent repeatedly cycles between a small set of tasks, and it is also where ARROW's advantage over DreamerV3 is most pronounced: maximum forgetting (Max-F) is small on Atari and essentially zero on CoinRun, well below DreamerV3 on both suites. ARROW also consistently exhibited strong recovery, which we hypothesize reflects superior representations learned through implicit multi-task learning; an alternative possibility is that the per-task budget in the two-cycle approach is sometimes too limited to fully learn a task in the first cycle, so that improvements at the second exposure partly reflect ordinary additional training rather than transfer.

Regarding sample efficiency, the picture is order-dependent and the threshold-crossing metric understates ARROW's advantage on Atari. Without shared structure, DreamerV3 records earlier threshold-crossings in the default and two-cycle orderings while ARROW does so in the reversed order (Tab. A.7); but the metric captures peak performance within a task's own training window, and the curves themselves (Fig. 3) show DreamerV3's peaks collapsing to near zero the moment the task changes, while ARROW's lower per-task peaks are sustained across the rest of training. Read against the curves rather than the threshold, ARROW is broader and more consistent. With shared structure (CoinRun), the picture is order-dependent: in the default order ARROW and DreamerV3 reach similar ceilings and the curves overlap, so DreamerV3's earlier threshold-crossing (Tab. A.8) does not reflect a sustained separation; in the reversed and two-cycle orders ARROW reaches a higher ceiling and is more consistent in approaching it. In both regimes the apparent per-task gap is the direct cost of ARROW's retention mechanism. Its long-term buffer mixes prior-task transitions into every update rather than concentrating them on the current task, paid in within-task peak performance and recovered in retention across tasks.

Our methods can be used in conjunction with prior state-of-the-art approaches to combating catastrophic forgetting such as EWC and P&C (Kirkpatrick et al., 2017; Schwarz et al., 2018) which work over network parameters, and CLEAR (Rolnick et al., 2019) which uses replay but typically operates over model-free approaches and uses behavior cloning of the policy from environment input to action output.

**Generalization.** Forward and backward transfer was significantly better for tasks with shared structure than those without shared structure; where the World Model's ability to generalize across tasks is very beneficial. Our experiments probe transfer across perturbations of a shared base environment (CoinRun visual variants), which involves both representation reuse and some skill adaptation; they do not constitute a graded difficulty curriculum in which simpler skills compose into more complex ones. Our two suites bracket the regimes but neither *is* a natural difficulty curriculum, and a domain-matched curriculum would be a more direct probe of forward transfer; we flag this as future work below.

**Buffer-ratio sensitivity.** The ablation in Sec. 5.3 traces a stability–plasticity axis: on Atari (no shared structure), shifting capacity from FIFO toward LTDM (from AR25 to AR75) reduces forgetting but degrades forward transfer, with AR50 sitting between the two extremes. On CoinRun (shared structure), forgetting is already low across all three variants and most pairwise gaps fall within the seed-to-seed IQRs; consistent with the intuition that plasticity costs little when tasks share representations, and indeed AR25 leads on most metrics in the default order. No variant dominates every setting within either suite, and the typical gap between variants is of the same order as seed-to-seed variability. We read this as evidence that the augmented-replay mechanism, rather than the specific FIFO/LTDM ratio, is the central design choice: the 50/50 split used in the main experiments is a robust default rather than a tuned optimum, and a workload-aware allocation (see Limitations below) is a natural extension once the regime (shared-structure vs. no-shared-structure) is known.

**Memory capacity.** As RL algorithms often consume high compute, we emphasize the benefit of lower computational and memory costs. ARROW does not scale up the buffer when compared to DreamerV3, but instead splits the available memory and intelligently uses past experience.

**Limitations.** ARROW currently allocates a fixed 50/50 split capacity to short-term and long-term buffers. We ablate this choice (25/75, 50/50, 75/25 splits) at fixed total budget on Atari and CoinRun in Sec. 5.3

(with per-metric breakdowns in Appendix A.1); as discussed under *Buffer-ratio sensitivity* above, no fixed ratio dominates across regimes, so a natural extension is to dynamically allocate memory based on task characteristics (e.g., adapting the FIFO/LTDM split when the data distribution shifts). A complementary stress test, varying the *total* buffer size while holding the FIFO/LTDM ratio fixed, is a separate question that we leave to future work. A second limitation concerns the per-task reward scaling described in Sec. 3.4: the static, linear, per-task scaling keeps the cross-task comparison interpretable and avoids requiring task IDs at training time, but it does require offline calibration on per-task baselines. Practitioners applying ARROW to unscaled task suites should expect this to be a critical hyperparameter; an adaptive scheme such as PopArt-style normalization (van Hasselt et al., 2016; Hessel et al., 2019) would be a natural drop-in replacement when offline calibration is impractical. A third, more fundamental limitation is the finite capacity inherent to any buffer-based method: as more tasks are explored and previous tasks are not revisited, an increasing number of samples from previous tasks will inevitably be lost, leading to forgetting. Fourth, our coverage of task orderings is limited: as the specific task ordering can result in significant differences in CL performance (Appendix G in (Rahimi-Kalahroudi et al., 2023)), we implemented two randomly chosen task sequences. This approach allowed us to quantify ordering effects while limiting computational cost. A follow-up study could dedicate more time to different permutations to better understand the relationship between individual environments and to shed some light on the unusually high performance of Frostbite in DV3.

**Future work.** Extending ARROW to continuous control or robotics domains like MuJoCo (Todorov et al., 2012) could further validate the generalization capabilities of our approach. Two further sweeps would strengthen the empirical picture but exceed our compute budget for this submission. First, varying the per-task time budget: we expect ARROW to remain relatively stable across this axis while DreamerV3 suffers at both extremes: longer per-task budgets lead to hyper-specialization and worse forgetting, while shorter budgets give insufficient learning of the current task and weaker transfer. Second, natural curricula such as graded Procgen difficulty levels, as discussed above; we predict ARROW's task-ID-free, distribution-matching long-term buffer to most clearly outperform DreamerV3 in this setting, with prior skills acting as compositional foundations for later tasks rather than merely being retained. Additionally, the experiments could be expanded by bringing ARROW to other model-based RL algorithms, e.g. TD-MPC (Hansen et al., 2022; 2024), or by combining it with existing techniques such as behavior cloning in CLEAR (Rolnick et al., 2019).

## 7 Conclusion

We extended the DreamerV3 World Model architecture with an augmented replay buffer (ARROW) and studied continual RL in two scenarios: tasks with and without shared structure. ARROW, DreamerV3, and TES-SAC were compared using the same memory budget (same-sized buffers). We evaluated forgetting, forward and backward transfer, and stability–plasticity metrics; ARROW's augmented replay buffer yielded substantial improvements in tasks without shared structure and a minor benefit in tasks with shared structure. The most practically relevant outcome is the two-cycle setting, the configuration that most closely matches a common continual-RL deployment scenario, where ARROW achieves essentially zero maximum forgetting between exposures on both suites, while DreamerV3 does not. Overall, these results support model-based RL with a World Model and a memory-efficient replay buffer as an effective and practical approach to continual RL, motivating future work.

### Acknowledgments

Abdallah Al Siyabi received support from a Monash Biomedicine Postgraduate Discovery Scholarship (Mbio). Markus R. Ernst was supported by an University International Postgraduate Award (UIPA) provided by UNSW Sydney. This research utilised Monash eResearch capabilities, including M3. All emojis used in the Figures are from OpenMoji – the open-source emoji and icon project. License: CC BY-SA 4.0.

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

# A   Tabular data & additional results

| Variant | Procgen flag | Description |
|---|---|---|
| Coinrun | — | regularly rendered game |
| +NB | `use_backgrounds = False` | removes decorative backgrounds |
| +RT | `restrict_themes = True` | restricts the set of level themes |
| +GA | `use_generated_assets = True` | enables procedurally generated assets |
| +MA | `use_monochrome_assets = True` | enables monochrome assets |
| +CA | `center_agent = False` | camera does not remain centered on the agent |

Table A.1: CoinRun task variations (Procgen configuration flags).

| Item | Specification |
|---|---|
| Training budget | 8.84 million environment frames over 540 epochs |
| Frame definition | CoinRun: 1 env. step per frame; Atari: 4 env. steps per frame |
| Default / Reversed schedule | 90 epochs per task, then shift to next task (single pass over 540 epochs) |
| Two-cycle schedule | Task shift every 45 epochs; cycle 1 ends at 270 epochs; cycle 2 uses remaining 270 epochs |
| Evaluation frequency | Every 10 epochs (55 checkpoints over 540 epochs, including epoch 0) |
| Evaluation scope | All tasks in the current sequence at each checkpoint |
| Policy for evaluation | Stochastic policy (random policy at epoch 0, trained policy thereafter) |
| Rollouts per task | CoinRun: 256; Atari: 16 |
| Return computation | Identify episode boundaries from reset and continuation flags; sum rewards within each episode |
| Reported statistics | Mean and standard deviation of episode returns per task |

Table A.2: Training schedule and evaluation protocol.

## A.1   Buffer-ratio ablation

The setup, summary, and numerical tables for the ablation experiments (AR25, AR50, and AR75) are reported in the main text (Sec. 5.3). This appendix collects the per-metric breakdowns with numerical tables (Tab. A.5, Tab. A.6) and bar charts (Fig. A.1, Fig. A.2).

## A.2   Single task sample efficiency

Tab. A.11 presents sample efficiency for individual Atari games learned in isolation. The results reveal heterogeneous task difficulty. In all comparisons where both model-based baselines reach the task threshold ARROW proves to be more sample efficient.

These single-task results establish important context: both ARROW and DreamerV3 struggle with certain games even in isolation, indicating that continual learning performance differences reflect both catastrophic forgetting and inherent task difficulty. The complementary failure modes (ARROW on Ms. Pac-Man/Seaquest, DreamerV3 on Frostbite) suggest different biases in how each method interacts with the game characteristics. The TES-SAC baseline fails to reach the 85 % threshold in every single Atari task.

Tab. A.12 presents single-task results for all six CoinRun variants. Both ARROW and DreamerV3 successfully reach the 85 % threshold on all variants. However, certain variants seem to be more difficult than others with steps ranging from about 500k (basic CoinRun) to 1.1M (+NB+RT+GA). ARROW outperforms DreamerV3 in all but two variants (+NB+RT+GA and +NB+RT+GA+MA+CA). Interestingly, for the particular variants and the infrequent cases where the TES-SAC algorithm successfully reaches the 85 % threshold it can be considered competitive with regard to sample efficiency.

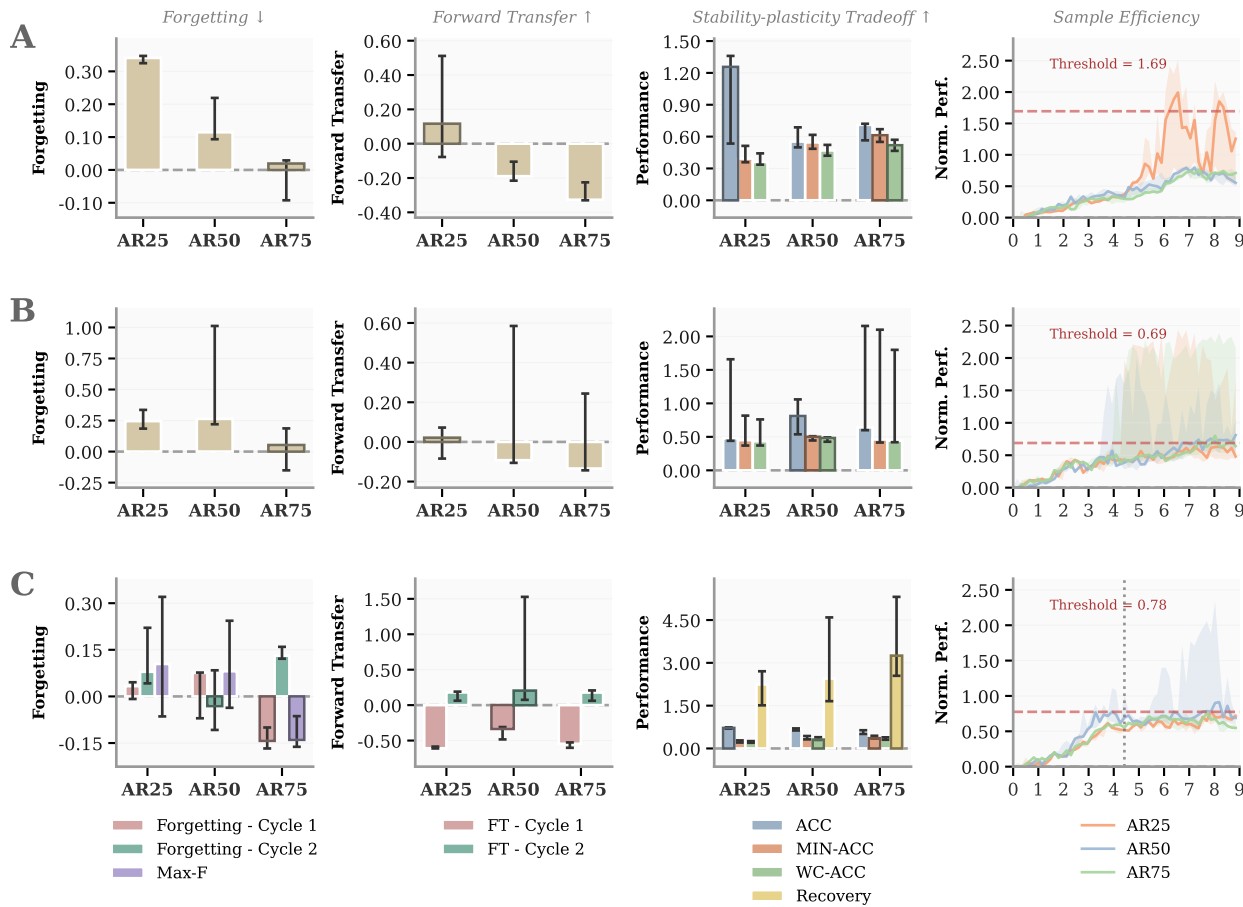

Figure A.1: **Atari** metrics for the ARROW buffer-ratio variants AR25, AR50, and AR75, shown as median with (0.25 - 0.75) quartile confidence intervals across 5 seeds, and calculated using normalized scores (Eq. 1). (A) Default task order (one-cycle). (B) Reversed task order (one-cycle). (C) Default task order (two-cycle).

## A.3 Validation of DreamerV3 implementation

To validate that our implementation faithfully represents DreamerV3, we compared it with the author's open-source implementation https://github.com/danijar/dreamerv3. Both implementations were evaluated on four tasks without shared structure. Fig. A.3 shows a side-by-side comparison, with our implementation on the left and the author's implementation on the right.

## A.4 Single-task runs

The parameters used for single-tasks for CoinRun (shared structure) are shown in Tab. A.13 and for Atari (without shared structure) in Tab. A.14.

The single-task results and the reward scales for Atari (without shared structure) are shown in Tab. A.15. The single-task results for CoinRun (shared structure) are shown in Tab. A.16.

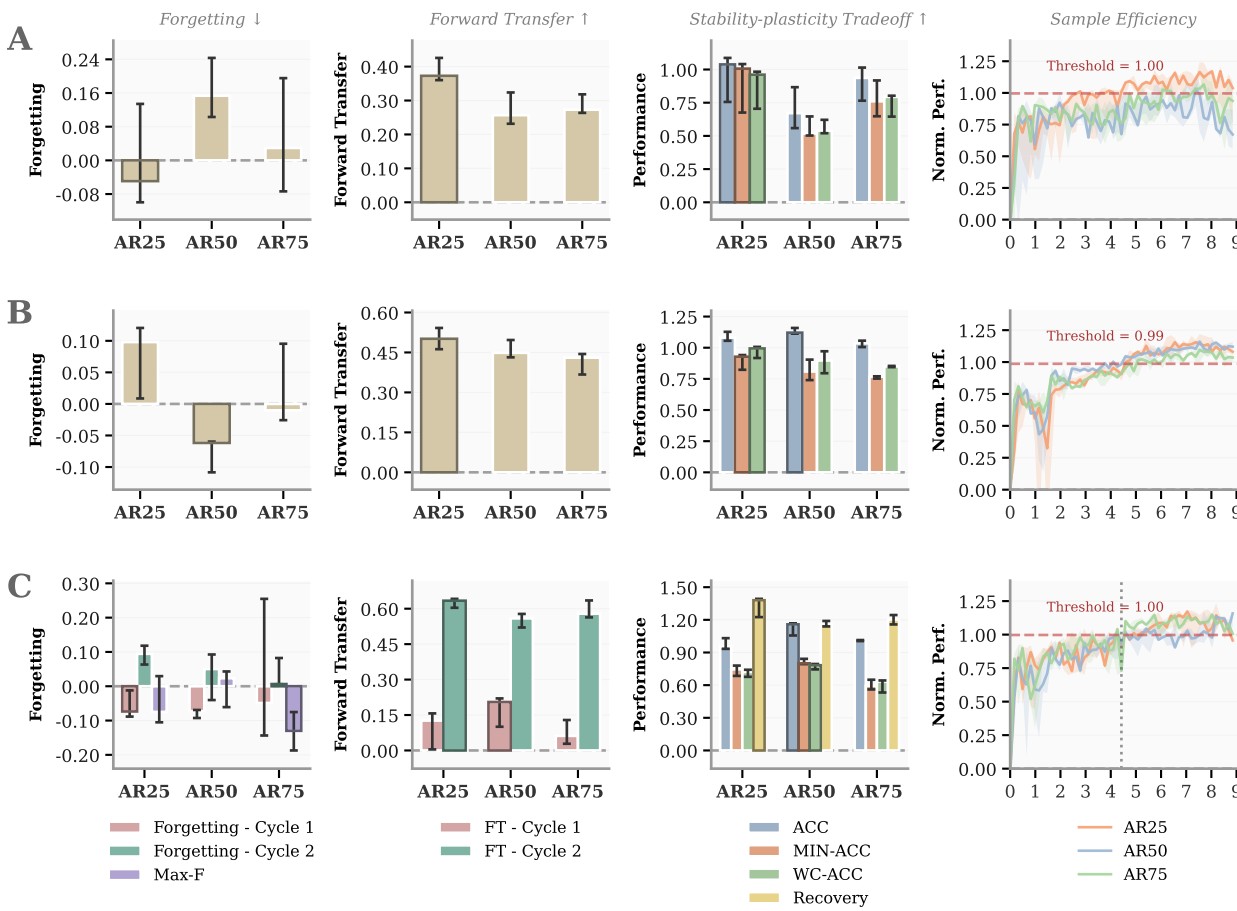

Figure A.2: **CoinRun** metrics for the ARROW buffer-ratio variants AR25, AR50, and AR75, shown as median with (0.25 - 0.75) quartile confidence intervals across 5 seeds, and calculated using normalized scores (Eq. 1). (A) Default order of tasks (one-cycle). (B) Reversed order of tasks (one-cycle). (C) Default order of tasks (two-cycle).

---

**Algorithm 1** ARROW training algorithm

---

**Hyperparameters:** World Model training iterations $K$.
**Input:** World Model $M$, augmented replay buffer $\mathcal{D}$, sequence of tasks $\boldsymbol{\mathcal{T}}_{1:T} = (\tau_1, \tau_2, \dots, \tau_T)$.
**for** $\tau = \tau_1, \tau_2, \dots, \tau_T$ **do**
    **for** $i = 1, 2, \dots, K$ **do**
        Train World Model $M$ on $\mathcal{D}$.
        Train actor $\pi$ using $M$.
        Use $\pi$ in $\tau$ and append episodes to $\mathcal{D}$.
    **end for**
**end for**

---

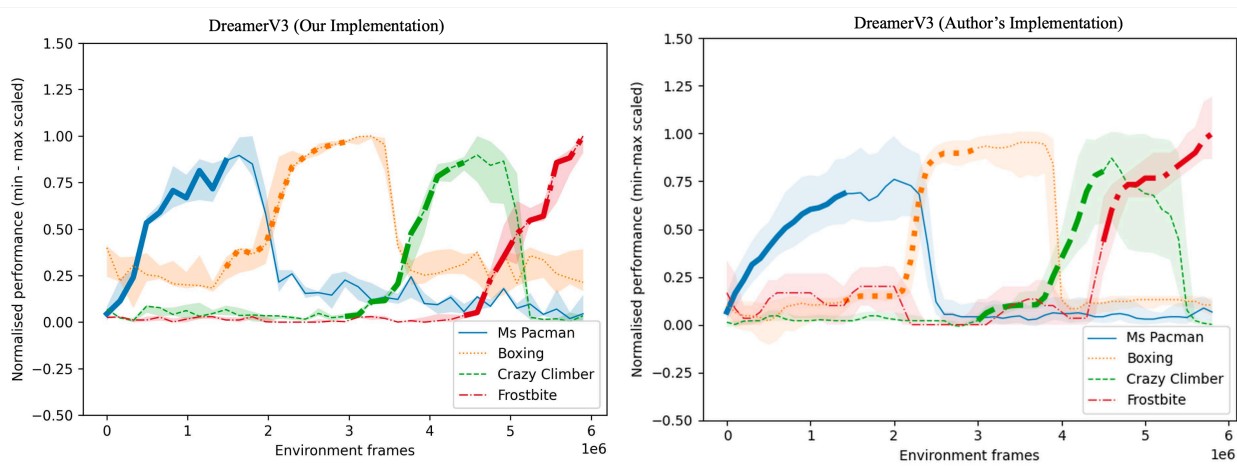

Figure A.3: Performance of DreamerV3, with bold line segments denoting the periods in which certain tasks are being trained. Scores are normalized using min-max normalization. The line is the median and the shaded area is between the 0.25 and 0.75 quantiles, of 5 seeds.

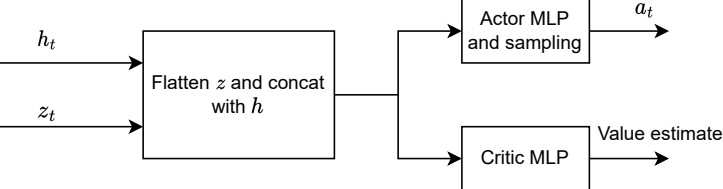

Figure B.1: Actor critic definition.

## B  World Models

### B.1  Training algorithm

### B.2  Network architecture

The Actor Critic architecture is shown in Fig. B.1. We adhered to most of the parameters and architectural choices of DreamerV3. Changes were primarily made to benefit wall time, as running continual learning experiments is computationally expensive. See Tab. C.1.

**CNN encoder and decoder**  Following DreamerV3, the convolutional feature extractor's input is a $64 \times 64$ RGB image as a resized environment frame. The encoder convolutional neural network (CNN) (LeCun et al., 1989) consists of stride 2 convolutions of doubling depth with the same padding until the image is at a resolution of $4 \times 4$, where it is flattened. We elected to use the "small" configuration of the hyperparameters controlling network architecture from DreamerV3 to appropriately manage experiment wall time. Hence, 4 convolutional layers were used with depths of 32, 64, 128, and 256 respectively. As with DreamerV3, we used channel-wise layer normalization (Ba et al., 2016) and SiLU (Hendrycks & Gimpel, 2016) activation for the CNN. The CNN decoder performs a linear transformation of the model state to a $4 \times 4 \times 256 = 4096$ vector before reshaping to a $4 \times 4$ image and inverting the encoder architecture to reconstruct the original environment frame.

**MLP**  All multi-layer perceptrons (MLP) within the RSSM, actor, and critic are 2 layers and 512 hidden units in accordance with the "small" configuration of DreamerV3.

### B.3    Augmented replay buffer

---

**Algorithm 2** Sampling from combined buffers $\mathcal{D}_1$ and $\mathcal{D}_2$

---

Combined (augmented) buffer $\mathcal{D} \doteq \{\mathcal{D}_1, \mathcal{D}_2\}$.
Uniformly sample $i \in \{1, 2\}$.
**return** Sampled minibatch from $\mathcal{D}_i$.

---

## C    Experimental Details

### C.1    Experimental parameters and execution time

General training and experimental parameters are provided in Tab. C.2. TES-SAC hyperparameters provided in Tab. C.3. The experimental run breakdown and wall-clock times are detailed in Tab. C.4 and Tab. C.5 respectively.

**Default task order**

| Method | Forgetting ↓ | FT ↑ | ACC ↑ | min-ACC ↑ | WC-ACC ↑ |
|---|---|---|---|---|---|
| ARROW | **0.115 [0.093 – 0.219]** | -0.190 [-0.216 – -0.105] | **0.553 [0.498 – 0.687]** | **0.546 [0.485 – 0.616]** | **0.470 [0.420 – 0.522]** |
| DV3 | 2.263 [2.062 – 2.732] | **0.612 [0.285 – 0.813]** | -0.029 [-0.046 – 0.131] | -0.249 [-0.265 – -0.181] | -0.141 [-0.152 – -0.128] |
| TES-SAC | 0.192 [0.151 – 0.213] | -0.273 [-0.589 – -0.234] | -0.010 [-0.036 – 0.044] | -0.144 [-0.148 – -0.124] | -0.114 [-0.119 – -0.096] |

**Reversed task order**

| Method | Forgetting ↓ | FT ↑ | ACC ↑ | min-ACC ↑ | WC-ACC ↑ |
|---|---|---|---|---|---|
| ARROW | 0.264 [0.220 – 1.012] | **-0.092 [-0.106 – 0.585]** | **0.811 [0.538 – 1.059]** | **0.500 [0.446 – 0.509]** | **0.482 [0.427 – 0.493]** |
| DV3 | 0.893 [0.744 – 0.983] | -0.299 [-0.333 – -0.198] | 0.012 [-0.001 – 0.091] | -0.222 [-0.235 – -0.204] | -0.049 [-0.090 – -0.036] |
| TES-SAC | **0.217 [0.214 – 0.233]** | -0.314 [-0.375 – -0.119] | -0.014 [-0.050 – 0.014] | -0.176 [-0.183 – -0.132] | -0.115 [-0.124 – -0.105] |

**Two-cycle training**

| Method | C1-F ↓ | C2-F ↓ | Max-F ↓ |
|---|---|---|---|
| ARROW | **0.076 [-0.070 – 0.077]** | **-0.031 [-0.108 – 0.084]** | **0.081 [-0.037 – 0.244]** |
| DV3 | 0.793 [0.599 – 1.007] | 0.718 [0.577 – 0.873] | 1.121 [0.899 – 1.177] |
| TES-SAC | 0.191 [0.144 – 0.195] | 0.136 [0.115 – 0.141] | 0.133 [0.083 – 0.179] |

| Method | C1-FT ↑ | C2-FT ↑ | Recovery ↑ |
|---|---|---|---|
| ARROW | **-0.335 [-0.484 – -0.309]** | **0.204 [0.075 – 1.528]** | **2.450 [1.657 – 4.592]** |
| DV3 | -0.342 [-0.487 – -0.246] | -0.545 [-0.610 – -0.449] | 0.729 [0.673 – 0.798] |
| TES-SAC | -0.587 [-0.605 – -0.560] | -0.359 [-0.472 – -0.263] | 0.140 [-2.060 – 0.401] |

| Method | ACC ↑ | min-ACC ↑ | WC-ACC ↑ |
|---|---|---|---|
| ARROW | **0.689 [0.630 – 0.703]** | **0.356 [0.319 – 0.436]** | **0.309 [0.300 – 0.394]** |
| DV3 | 0.014 [-0.008 – 0.093] | -0.294 [-0.364 – -0.236] | -0.226 [-0.287 – -0.174] |
| TES-SAC | 0.038 [0.025 – 0.101] | -0.165 [-0.222 – -0.160] | -0.137 [-0.181 – -0.132] |

Table A.3: **Atari** metrics (Baselines) using median [IQR] across seeds for (A, B and C). Best performance is in bold.

**Default task order**

| Method | Forgetting ↓ | FT ↑ | ACC ↑ | min-ACC ↑ | WC-ACC ↑ |
|---|---|---|---|---|---|
| ARROW | 0.154 [0.103 − 0.243] | 0.257 [0.232 − 0.324] | 0.672 [0.559 − 0.867] | **0.518 [0.503 − 0.647]** | **0.539 [0.520 − 0.621]** |
| DV3 | 0.416 [0.371 − 0.541] | **0.503 [0.455 − 0.535]** | **0.847 [0.753 − 0.862]** | 0.083 [-0.597 − 0.247] | 0.263 [-0.331 − 0.366] |
| TES-SAC | **0.002 [-0.053 − 0.009]** | 0.495 [0.424 − 0.501] | 0.595 [0.587 − 0.753] | 0.434 [0.432 − 0.439] | 0.510 [0.493 − 0.523] |

**Reversed task order**

| Method | Forgetting ↓ | FT ↑ | ACC ↑ | min-ACC ↑ | WC-ACC ↑ |
|---|---|---|---|---|---|
| ARROW | **-0.062 [-0.108 − -0.060]** | 0.449 [0.432 − 0.497] | **1.121 [1.113 − 1.159]** | **0.809 [0.740 − 0.904]** | **0.898 [0.795 − 0.972]** |
| DV3 | 0.340 [0.197 − 0.482] | **0.578 [0.556 − 0.591]** | 0.888 [0.687 − 1.079] | 0.603 [0.498 − 0.706] | 0.689 [0.625 − 0.806] |
| TES-SAC | -0.055 [-0.071 − -0.049] | 0.406 [0.259 − 0.436] | 0.660 [0.649 − 0.693] | 0.468 [0.448 − 0.470] | 0.504 [0.500 − 0.525] |

**Two-cycle training**

| Method | C1-F ↓ | C2-F ↓ | Max-F ↓ |
|---|---|---|---|
| ARROW | **-0.073 [-0.093 − -0.069]** | 0.050 [-0.040 − 0.093] | 0.023 [-0.061 − 0.043] |
| DV3 | 0.472 [0.108 − 0.883] | 0.296 [0.294 − 0.522] | 0.184 [0.126 − 0.441] |
| TES-SAC | 0.044 [0.027 − 0.057] | **-0.016 [-0.050 − 0.001]** | **-0.010 [-0.106 − 0.013]** |

| | C1-FT ↑ | C2-FT ↑ | Recovery ↑ |
|---|---|---|---|
| ARROW | 0.205 [0.100 − 0.220] | 0.559 [0.520 − 0.578] | 1.164 [1.140 − 1.190] |
| DV3 | 0.230 [0.187 − 0.238] | 0.571 [0.378 − 0.614] | **1.207 [1.092 − 1.551]** |
| TES-SAC | **0.359 [0.313 − 0.501]** | **0.596 [0.559 − 0.849]** | 1.125 [0.988 − 1.228] |

| | ACC ↑ | min-ACC ↑ | WC-ACC ↑ |
|---|---|---|---|
| ARROW | **1.158 [1.056 − 1.168]** | **0.800 [0.793 − 0.842]** | **0.783 [0.745 − 0.794]** |
| DV3 | 0.535 [0.480 − 0.571] | -0.126 [-0.181 − 0.111] | 0.041 [-0.041 − 0.217] |
| TES-SAC | 0.758 [0.681 − 0.818] | 0.414 [0.286 − 0.492] | 0.516 [0.360 − 0.589] |

Table A.4: **Coinrun** metrics (Baselines) using median [IQR] across seeds for (A, B and C). Best performance is in bold.

**Default task order**

| Method | Forgetting ↓ | FT ↑ | ACC ↑ | min-ACC ↑ | WC-ACC ↑ |
|---|---|---|---|---|---|
| AR25 | 0.341 [0.325 − 0.347] | **-0.087** [**-0.164** − **0.020**] | **1.256** [**0.534** − **1.360**] | 0.393 [0.358 − 0.513] | 0.361 [0.336 − 0.442] |
| AR50 | 0.115 [0.093 − 0.219] | -0.190 [-0.216 − -0.105] | 0.553 [0.498 − 0.687] | 0.546 [0.485 − 0.616] | 0.470 [0.420 − 0.522] |
| AR75 | **0.019** [**-0.092** − **0.029**] | -0.470 [-0.484 − -0.424] | 0.711 [0.565 − 0.721] | **0.612** [**0.550** − **0.670**] | **0.518** [**0.466** − **0.570**] |

**Reversed task order**

| Method | Forgetting ↓ | FT ↑ | ACC ↑ | min-ACC ↑ | WC-ACC ↑ |
|---|---|---|---|---|---|
| AR25 | 0.245 [0.185 − 0.336] | **-0.061** [**-0.153** − **-0.028**] | 0.481 [0.441 − 1.659] | 0.451 [0.370 − 0.817] | 0.434 [0.371 − 0.760] |
| AR50 | 0.264 [0.220 − 1.012] | -0.092 [-0.106 − 0.585] | **0.811** [**0.538** − **1.059**] | **0.500** [**0.446** − **0.509**] | **0.482** [**0.427** − **0.493**] |
| AR75 | **0.053** [**-0.151** − **0.187**] | -0.328 [-0.328 − -0.283] | 0.641 [0.599 − 2.156] | 0.462 [0.415 − 2.100] | 0.444 [0.420 − 1.799] |

**Two-cycle training**

| Method | C1-F ↓ | C2-F ↓ | Max-F ↓ | Recovery ↑ |
|---|---|---|---|---|
| AR25 | 0.033 [-0.008 − 0.046] | 0.079 [0.042 − 0.221] | 0.105 [-0.064 − 0.320] | 2.243 [1.512 − 2.705] |
| AR50 | 0.076 [-0.070 − 0.077] | **-0.031** [**-0.108** − **0.084**] | 0.081 [-0.037 − 0.244] | 2.450 [1.657 − 4.592] |
| AR75 | **-0.143** [**-0.167** − **-0.100**] | 0.131 [0.121 − 0.160] | **-0.140** [**-0.162** − **-0.063**] | **3.252** [**2.547** − **5.312**] |

| Method | C1-FT ↑ | C2-FT ↑ | min-ACC ↑ | WC-ACC ↑ |
|---|---|---|---|---|
| AR25 | -0.652 [-0.671 − -0.643] | 0.063 [-0.035 − 0.077] | 0.214 [0.206 − 0.282] | 0.215 [0.200 − 0.257] |
| AR50 | **-0.335** [**-0.484** − **-0.309**] | **0.204** [**0.075** − **1.528**] | 0.356 [0.319 − 0.436] | **0.309** [**0.300** − **0.394**] |
| AR75 | -0.658 [-0.712 − -0.648] | -0.073 [-0.156 − -0.050] | **0.359** [**0.351** − **0.444**] | 0.307 [0.305 − 0.390] |

| Method | ACC ↑ |
|---|---|
| AR25 | **0.718** [**0.693** − **0.740**] |
| AR50 | 0.689 [0.630 − 0.703] |
| AR75 | 0.548 [0.510 − 0.638] |

Table A.5: **ARROW** Buffer-ratio ablation on **Atari** metrics using median [IQR] across seeds for (A, B and C). Best performance is in bold.

**Default task order**

| Method | Forgetting ↓ | FT ↑ | ACC ↑ | min-ACC ↑ | WC-ACC ↑ |
|---|---|---|---|---|---|
| AR25 | **-0.049 [-0.100 − 0.134]** | **0.314 [0.310 − 0.371]** | **1.038 [0.756 − 1.087]** | **1.007 [0.676 − 1.041]** | **0.961 [0.705 − 0.984]** |
| AR50 | 0.154 [0.103 − 0.243] | 0.257 [0.232 − 0.324] | 0.672 [0.559 − 0.867] | 0.518 [0.503 − 0.647] | 0.539 [0.520 − 0.621] |
| AR75 | 0.030 [-0.074 − 0.195] | 0.256 [0.246 − 0.299] | 0.937 [0.765 − 1.015] | 0.761 [0.648 − 0.918] | 0.794 [0.645 − 0.803] |

**Reversed task order**

| Method | Forgetting ↓ | FT ↑ | ACC ↑ | min-ACC ↑ | WC-ACC ↑ |
|---|---|---|---|---|---|
| AR25 | 0.098 [0.009 − 0.120] | 0.424 [0.386 − 0.467] | 1.081 [1.056 − 1.128] | **0.928 [0.823 − 0.941]** | **0.994 [0.917 − 1.007]** |
| AR50 | **-0.062 [-0.108 − -0.060]** | **0.449 [0.432 − 0.497]** | **1.121 [1.113 − 1.159]** | 0.809 [0.740 − 0.904] | 0.898 [0.795 − 0.972] |
| AR75 | -0.010 [-0.026 − 0.095] | 0.422 [0.331 − 0.428] | 1.035 [1.005 − 1.056] | 0.768 [0.752 − 0.771] | 0.851 [0.844 − 0.854] |

**Two-cycle training**

| Method | C1-F ↓ | C2-F ↓ | Max-F ↓ |
|---|---|---|---|
| AR25 | **-0.074 [-0.088 − -0.012]** | 0.095 [0.063 − 0.118] | -0.075 [-0.105 − 0.029] |
| AR50 | -0.073 [-0.093 − -0.069] | 0.050 [-0.040 − 0.093] | 0.023 [-0.061 − 0.043] |
| AR75 | -0.049 [-0.144 − 0.255] | **0.006 [0.006 − 0.082]** | **-0.130 [-0.187 − -0.075]** |

| | C1-FT ↑ | C2-FT ↑ | Recovery ↑ |
|---|---|---|---|
| AR25 | 0.080 [-0.050 − 0.111] | **0.560 [0.532 − 0.568]** | **1.380 [1.225 − 1.393]** |
| AR50 | **0.205 [0.100 − 0.220]** | 0.559 [0.520 − 0.578] | 1.164 [1.140 − 1.190] |
| AR75 | 0.016 [-0.012 − 0.090] | 0.551 [0.550 − 0.601] | 1.203 [1.158 − 1.244] |

| | ACC ↑ | min-ACC ↑ | WC-ACC ↑ |
|---|---|---|---|
| AR25 | 0.957 [0.933 − 1.032] | 0.745 [0.685 − 0.780] | 0.722 [0.676 − 0.741] |
| AR50 | **1.158 [1.056 − 1.168]** | **0.800 [0.793 − 0.842]** | **0.783 [0.745 − 0.794]** |
| AR75 | 1.006 [1.004 − 1.014] | 0.594 [0.560 − 0.650] | 0.634 [0.532 − 0.643] |

Table A.6: **ARROW Buffer-ratio ablation on CoinRun** metrics using median [IQR] across seeds for (A, B and C). Best performance is in bold.

| 85% Threshold | Frame at Middle | Method | Max Perf. | Env. Frames (median [q25–q75]) | Runs ≥85% |
|---|---|---|---|---|---|
| | | **Default Task Order** | | | |
| 1.71 | 4,423,680 | ARROW | 0.80 | Never reached threshold | 0/5 |
| | | **DV3** | 2.01 | **5,652,480 [5,570,560 − 5,775,360]** | 4/5 |
| | | TES-SAC | 0.16 | Never reached threshold | 0/5 |
| | | **Reversed Task Order** | | | |
| 0.69 | 4,423,680 | **ARROW** | 0.81 | **3,604,480 [3,522,560 − 5,079,040]** | 3/5 |
| | | DV3 | 0.30 | Never reached threshold | 0/5 |
| | | TES-SAC | 0.15 | Never reached threshold | 0/5 |
| | | **Two-Cycle Training** | | | |
| 0.99 | 4,423,680 | ARROW | 0.91 | 3,440,640 [3,194,880 − 3,522,560] | 3/5 |
| | | **DV3** | 1.17 | **3,112,960 [2,867,200 − 3,112,960]** | 3/5 |
| | | TES-SAC | 0.18 | Never reached threshold | 0/5 |

Table A.7: Atari (Normalized) continual learning sample-efficiency. Best performance is written in bold.

| 85% Threshold | Frame at Middle | Method | Max Perf. | Env. Frames (median [q25–q75]) | Runs ≥85% |
|---|---|---|---|---|---|
| | | **Default Task Order** | | | |
| 0.88 | 4,423,680 | ARROW | 1.03 | 1,638,400 [655,360 − 2,293,760] | 5/5 |
| | | **DV3** | 1.04 | **491,520 [327,680 − 819,200]** | 5/5 |
| | | TES-SAC | 0.75 | Never reached threshold | 0/5 |
| | | **Reversed Task Order** | | | |
| 0.99 | 4,423,680 | **ARROW** | 1.16 | **3,604,480 [3,112,960 − 3,768,320]** | 5/5 |
| | | DV3 | 1.11 | 3,604,480 [3,276,800 − 4,751,360] | 5/5 |
| | | TES-SAC | 0.74 | Never reached threshold | 0/5 |
| | | **Two-Cycle Training** | | | |
| 0.98 | 4,423,680 | ARROW | 1.16 | 3,768,320 [2,949,120 − 3,932,160] | 5/5 |
| | | **DV3** | 1.07 | **1,802,240 [1,474,560 − 1,802,240]** | 5/5 |
| | | TES-SAC | 0.78 | Never reached threshold | 0/5 |

Table A.8: CoinRun (Normalized) continual learning sample-efficiency. Best performance is written in bold.

| 85% Threshold | Frame at Middle | Method | Max Perf. | Env. Frames (median [q25–q75]) | Runs ≥85% |
|---|---|---|---|---|---|
| | | **Default Task Order** | | | |
| 1.69 | 4,423,680 | **AR25** | 1.99 | **5,734,400 [5,488,640 − 5,898,240]** | 3/5 |
| | | AR50 | 0.80 | Never reached threshold | 0/5 |
| | | AR75 | 0.77 | Never reached threshold | 0/5 |
| | | **Reversed Task Order** | | | |
| 0.69 | 4,423,680 | AR25 | 0.64 | 3,686,400 [3,563,520 − 3,809,280] | 2/5 |
| | | **AR50** | 0.81 | **3,604,480 [3,522,560 − 5,079,040]** | 3/5 |
| | | AR75 | 0.80 | 4,096,000 [3,768,320 − 5,734,400] | 3/5 |
| | | **Two-Cycle Training** | | | |
| 0.78 | 4,423,680 | AR25 | 0.79 | 7,700,480 [7,004,160 − 7,864,320] | 4/5 |
| | | **AR50** | 0.91 | **2,949,120 [2,785,280 − 3,932,160]** | 5/5 |
| | | AR75 | 0.76 | 6,389,760 [6,144,000 − 6,635,520] | 2/5 |

Table A.9: ARROW Buffer-ratio ablation on Atari - (Normalized) continual learning sample-efficiency. Best performance is written in bold.

| 85% Threshold | Frame at Middle | Method | Max Perf. | Env. Frames (median [q25–q75]) | Runs ≥85% |
|---|---|---|---|---|---|
| | | **Default Task Order** | | | |
| 1.00 | 4,423,680 | **AR25** | 1.17 | **2,621,440 [2,457,600 − 4,751,360]** | 5/5 |
| | | AR50 | 1.03 | 4,014,080 [3,072,000 − 4,751,360] | 4/5 |
| | | AR75 | 1.07 | 4,751,360 [4,259,840 − 5,242,880] | 5/5 |
| | | **Reversed Task Order** | | | |
| 0.99 | 4,423,680 | AR25 | 1.16 | 4,587,520 [3,768,320 − 4,587,520] | 5/5 |
| | | **AR50** | 1.16 | **3,604,480 [3,112,960 − 3,768,320]** | 5/5 |
| | | AR75 | 1.10 | 4,751,360 [1,966,080 − 5,242,880] | 5/5 |
| | | **Two-Cycle Training** | | | |
| 1.00 | 4,423,680 | AR25 | 1.17 | 3,276,800 [2,129,920 − 4,259,840] | 5/5 |
| | | AR50 | 1.16 | 3,768,320 [2,949,120 − 3,932,160] | 5/5 |
| | | **AR75** | 1.15 | **2,457,600 [2,129,920 − 3,112,960]** | 5/5 |

Table A.10: ARROW Buffer-ratio ablation on CoinRun - (Normalized) continual learning sample-efficiency. Best performance is written in bold.

| Task | Task-Specific Threshold (85%) | Method | Max Perf. | Env. Frames (median [q25–q75]) | Runs ≥85% |
|---|---|---|---|---|---|
| Ms. Pac-Man | 131.43 | ARROW | 96.26 | Never reached 85% of peak | 0/5 |
| | | **DV3** | 154.62 | **1,310,720 [1,310,720 − 1,310,720]** | 1/5 |
| | | TES-SAC | 46.00 | Never reached 85% of peak | 0/5 |
| Boxing | 84.05 | **ARROW** | 98.88 | **819,200 [655,360 − 983,040]** | 5/5 |
| | | DV3 | 98.71 | 983,040 [819,200 − 983,040] | 5/5 |
| | | TES-SAC | 3.12 | Never reached 85% of peak | 0/5 |
| Crazy Climber | 109.27 | **ARROW** | 128.56 | **983,040 [983,040 − 983,040]** | 1/5 |
| | | DV3 | 118.88 | 1,228,800 [1,105,920 − 1,351,680] | 4/5 |
| | | TES-SAC | 12.31 | Never reached 85% of peak | 0/5 |
| Frostbite | 68.96 | **ARROW** | 81.12 | **1,474,560 [1,474,560 − 1,474,560]** | 1/5 |
| | | DV3 | 55.42 | Never reached 85% of peak | 0/5 |
| | | TES-SAC | 42.12 | Never reached 85% of peak | 0/5 |
| Seaquest | 533.18 | ARROW | 401.05 | Never reached 85% of peak | 0/5 |
| | | **DV3** | 627.27 | **1,310,720 [1,310,720 − 1,310,720]** | 1/5 |
| | | TES-SAC | 258.75 | Never reached 85% of peak | 0/5 |
| Enduro | 339.06 | **ARROW** | 398.89 | **1,146,880 [1,146,880 − 1,474,560]** | 5/5 |
| | | DV3 | 374.10 | 1,228,800 [1,146,880 − 1,351,680] | 4/5 |
| | | TES-SAC | 15.50 | Never reached 85% of peak | 0/5 |

Table A.11: Atari single-task sample-efficiency (raw rewards). Best performance is written in bold.

| Task | Task-Specific Threshold (85%) | Method | Max Perf. | Env. Frames (median [q25–q75]) | Runs ≥85% |
|---|---|---|---|---|---|
| CoinRun | 6.08 | **ARROW** | 6.99 | **491,520 [491,520 − 819,200]** | 5/5 |
| | | **DV3** | 7.15 | **491,520 [491,520 − 819,200]** | 5/5 |
| | | TES-SAC | 6.56 | 655,360 [655,360 − 655,360] | 2/5 |
| +NB | 6.84 | **ARROW** | 7.46 | **983,040 [901,120 − 983,040]** | 4/5 |
| | | DV3 | 8.05 | 1,146,880 [983,040 − 1,310,720] | 5/5 |
| | | TES-SAC | 6.45 | Never reached threshold | 0/5 |
| +NB+RT | 6.24 | ARROW | 7.34 | 819,200 [655,360 − 983,040] | 5/5 |
| | | DV3 | 7.30 | 983,040 [983,040 − 1,146,880] | 5/5 |
| | | **TES-SAC** | 6.52 | **163,840 [163,840 − 163,840]** | 1/5 |
| +NB+RT+GA | 7.04 | ARROW | 8.28 | 1,310,720 [1,146,880 − 1,392,640] | 3/5 |
| | | **DV3** | 8.05 | **1,146,880 [983,040 − 1,146,880]** | 5/5 |
| | | TES-SAC | 6.64 | Never reached threshold | 0/5 |
| +NB+RT+GA+MA | 6.97 | **ARROW** | 8.20 | **491,520 [491,520 − 983,040]** | 5/5 |
| | | DV3 | 7.89 | 983,040 [819,200 − 983,040] | 5/5 |
| | | TES-SAC | 6.02 | Never reached threshold | 0/5 |
| +NB+RT+GA+MA+CA | 5.45 | ARROW | 6.25 | 491,520 [491,520 − 655,360] | 5/5 |
| | | **DV3** | 6.41 | **491,520 [327,680 − 655,360]** | 5/5 |
| | | TES-SAC | 6.33 | 573,440 [532,480 − 614,400] | 2/5 |

Table A.12: CoinRun single-task sample-efficiency (raw rewards). Best performance is written in bold.

| | Env. frames | Env. steps | Replay buffer capacity |
|---|---|---|---|
| ARROW | 1.47M | 1.47M | $2 \times 512$ sequences $\times T{=}512$ ($2^{19}$ obs.) |
| DreamerV3 | 1.47M | 1.47M | 1024 sequences $\times T{=}512$ ($2^{19}$ obs.) |
| TES-SAC | 1.47M | 1.47M | 1024 sequences $\times T{=}512$ ($2^{19}$ obs.) |

Table A.13: CoinRun single-task training parameters.

| | Env. frames | Env. steps | Replay buffer capacity |
|---|---|---|---|
| ARROW | 1.47M | 5.89M | $2 \times 512$ sequences $\times T{=}512$ ($2^{19}$ obs.) |
| DreamerV3 | 1.47M | 5.89M | 1024 sequences $\times T{=}512$ ($2^{19}$ obs.) |
| TES-SAC | 1.47M | 5.89M | 1024 sequences $\times T{=}512$ ($2^{19}$ obs.) |

Table A.14: Atari single-task training parameters.

| Task | Reward scale | Random | ARROW |
|---|---|---|---|
| Ms. Pac-Man | 0.05 | 12.40 | 1540.30 |
| Boxing | 1 | 0.51 | 90.27 |
| Crazy Climber | 0.001 | 7.49 | 109245.16 |
| Frostbite | 0.2 | 14.48 | 297.83 |
| Seaquest | 0.5 | 38.47 | 439.62 |
| Enduro | 0.5 | 0.01 | 707.47 |

Table A.15: Atari single-task experimental results, median across 5 random seeds. Scores are unnormalized at the end of training.

| Task | Random | ARROW |
|---|---|---|
| CoinRun | 2.78 | 6.09 |
| +NB | 2.45 | 7.14 |
| +NB+RT | 2.70 | 6.89 |
| +NB+RT+GA | 2.62 | 6.85 |
| +NB+RT+GA+MA | 2.50 | 7.89 |
| +NB+RT+GA+MA+CA | 2.69 | 5.78 |

Table A.16: Procgen CoinRun single-task experimental results, median across 5 random seeds. Scores are unnormalized at the end of training.

| Name | ARROW | DreamerV3 | TES-SAC |
|---|---|---|---|
| Replay capacity (FIFO) | 0.26M | 0.52M | 0.52M |
| Replay capacity (long-term) | 0.26M | 0 | 0 |
| Batch size | 16 | 16 | 128 |
| Learning rate | $1 \times 10^{-4}$ | $1 \times 10^{-4}$ | $5 \times 10^{-4}$ |
| Activation (MLP) | LayerNorm+SiLU | LayerNorm+SiLU | ReLU |
| Activation (GRU) | tanh | tanh | − |
| GRU units | 512 | 512 | − |
| MLP features | 512 | 512 | 512 |
| MLP layers | 2 | 2 | 2 |
| CNN depth | 32 | 32 | 32 |

Table C.1: Hyperparameters.

| Parameter | Atari | CoinRun |
|---|---|---|
| **Data Collection** | | |
| Parallel environments | 4 | 4 |
| Sequence length | 4096 | 4096 |
| Environment repeat | 4 | 1 |
| Data sequences per batch | 32 | 32 |
| Max sequences in buffer | 512 | 512 |
| Sequence time steps | 512 | 512 |
| **Minibatch Configuration** | | |
| Minibatch time size | 32 | 32 |
| Minibatch number size | 16 | 16 |
| **Environment-Specific** | | |
| Action space size | 18 | 15 |
| Image size | $64 \times 64$ | $64 \times 64$ |

Table C.2: General training and experimental parameters.

| Parameter | Value |
|---|---|
| **Learning Rates** | |
| Policy learning rate | $5 \times 10^{-4}$ |
| Q-network learning rate | $5 \times 10^{-4}$ |
| Entropy coefficient (alpha) learning rate | $5 \times 10^{-4}$ |
| **Core SAC Hyperparameters** | |
| Batch size | 128 |
| Discount factor (gamma) | 0.99 |
| Soft target update (tau) | 0.005 |
| Initial entropy coefficient (alpha) | 0.2 |
| Target entropy | $0.8 \times \log(|\mathcal{A}|)$ |
| Gradient clipping | 5.0 |
| **Target Entropy Scheduling (TES-SAC)** | |
| TES lambda | 0.999 |
| Average threshold | 0.01 |
| Standard deviation threshold | 0.05 |
| Discount factor k | 0.98 |
| TES period T | 1000 |

Table C.3: TES-SAC Hyperparameters

| Benchmark | Task | Runs | Total |
|---|---|---|---|
| Atari | Single-task | 6 tasks × 5 seeds | 30 |
| | Continual learning | 3 settings (A, B, C) × 5 seeds | 15 |
| CoinRun | Single-task | 6 tasks × 5 seeds | 30 |
| | Continual learning | 3 settings (A, B, C) × 5 seeds | 15 |
| Per method, per benchmark | | (30 + 15) | 45 |
| Per benchmark, all methods (ARROW, DV3, TES-SAC) | | 45 runs × 3 methods | 135 |
| Total (Atari + CoinRun, all methods) | | 135 runs × 2 benchmarks | **270** |

Table C.4: Experimental run breakdown across benchmarks and methods.

All experiments were executed on a single NVIDIA A40 or A100 GPU (depending on availability).

| Method | Single-task wall time | Continual learning wall time |
|---|---|---|
| ARROW / DV3 | 6 hours | 1–2 days (CoinRun CL: ∼30 hours; Atari CL: ∼50 hours) |
| TES-SAC | 3 hours | 17 hours |

Table C.5: Wall-clock time per method and setting.

