# OpenReview forum: "ARROW: Augmented Replay for RObust World models"
_TMLR — Accepted by TMLR_

### Review · Reviewer_duq1 · 2026-03-15

**Summary Of Contributions:**

The authors present ARROW which they developed as a model-based continual reinforcement learning system that uses DreamerV3 as its foundation. ARROW uses two memory buffers to prevent catastrophic forgetting while it keeps memory use at optimal levels through its short-term FIFO buffer and its long-term global distribution matching (LTDM) buffer. The approach uses spliced rollouts to achieve task diversity while operating under memory constraints. The method is tested on two types of tasks which include non-shared structure tasks (Atari) and shared structure tasks (Procgen CoinRun).

The paper examines an important obstacle which exists in model-based continual reinforcement learning because traditional replay buffers require excessive memory resources. The evaluation framework tests its model-based DreamerV3 and model-free TES-SAC baselines through memory budget testing across all methods which serves as a matching constraint. The authors utilize a comprehensive suite of metrics, including forgetting, forward transfer, worst-case accuracy (WC-ACC), maximum forgetting (Max-F), and recovery through a two-cycle training schedule. ARROW shows a better capacity to prevent catastrophic forgetting during non-shared structure tasks than DreamerV3.

The model-free baseline (TES-SAC) fails to adequately learn the Atari tasks initially. The comparative stability/forgetting metrics become unhelpful because of this reason. The method requires static, linear reward transformations when tasks have vastly different reward scales, as ARROW struggles to balance learning across environments with heavily skewed returns. ARROW requires extra environment frames to accomplish its performance objectives because it sacrifices sample efficiency during shared structure tasks which require more time than DreamerV3. The 50/50 memory allocation split between the short-term and long-term buffers is arbitrary and unexplored.

**Additional Comments:**

The review process benefits from an anonymous link which provides access to the source code for verification purposes and this practice helps researchers to reproduce their results.

**Audience:**

Yes

**Audience Explanation:**

The stability-plasticity dilemma presents a challenge to continuous learning in reinforcement learning. World Models demonstrate potential for off-policy learning and ongoing usage yet their dependency on large replay buffers results in significant scalability problems. ARROW demonstrates that memory bottlenecks can be reduced through neuroscience-based strategic distribution-matching yet performance remains competitive which provides vital information for lifelong learning researchers and efficient RL experts and model-based architecture developers.

**Broader Impact Concerns:**

No major concerns. This research emphasizes basic methods of reinforcement learning and methods of representation learning. The study creates autonomous agents which operate in unpredictable environments yet their societal effects will not begin to emerge until their implementation occurs and thus the study does not require a Broader Impact Statement to address its ethical concerns.

**Claims And Evidence:**

Yes

**Claims Explanation:**

The authors establish a fair testing ground by ensuring all baselines operate under the exact same memory constraints. The researchers test various task configurations which include default ordering , reversed ordering , and a two-cycle split. The claims regarding ARROW's ability to retain previous knowledge are well-supported by the empirical results which show near-zero maximum forgetting (Max-F) in the Atari two-cycle setup. The authors reveal their limitations by discussing three specific areas which include task-order sensitivity , baseline failures , and reward scaling problems.

**Requested Changes:**

The TES-SAC system shows its main weakness through its performance drop in the Atari game collection. The comparison becomes invalid because TES-SAC does not meet baseline performance requirements which makes it impossible to demonstrate its "low forgetting" capability. I recommend either tuning TES-SAC so it achieves a baseline competency on Atari or introducing an alternative model-free baseline that actually learns the tasks, so the forgetting metrics hold comparative weight.

The study needs to present an ablation study which shows the effects of maintaining a fixed 50/50 capacity split between the FIFO and LTDM buffers. The model response to this ratio demonstration needs to show buffer dynamics systems operation.

The research needs to present a complete explanation about how reward scaling creates problems for its effectiveness. The method shows its failure when rewards vary by 100 times, which acts as a teaching example for practitioners who want to apply this method to unscaled task suites.

---

> ### Author Response · Authors · 2026-05-15
> **Response to reviewer duq1**
>
> ### 1: TES-SAC fails to learn Atari, so its low forgetting score is not informative
>
> We chose SAC as the strongest model-free baseline at this memory budget, using Target-Entropy Scheduled SAC (TES-SAC) (Xu et al., 2021) because its scheduled target entropy (higher exploration early, lower-entropy exploitation later) suits long curricula. Justification now in Sec. 4.2.
>
> The reviewer is correct that TES-SAC's low forgetting on Atari is an artefact of low absolute performance, a model that never learns has nothing to forget. A grid search at ARROW's budget over the main TES-SAC hyperparameters found no configuration that learns Atari reliably; we frame this as a limitation of model-free methods at this budget, not of TES-SAC itself. We retain TES-SAC because the asymmetry is diagnostic; forgetting must be read alongside ACC / WC-ACC. This is now explicit in Sec. 5.1 and Discussion.
>
> ### 2: Ablation of the 50/50 FIFO/LTDM buffer split
>
> We have run the requested ablation. Holding the total replay budget fixed, we trained ARROW on all settings with three FIFO/LTDM splits: AR25 (75/25), AR50 (50/50, original), AR75 (25/75). Setup, results, per-metric bar charts, and design implications appear in Sec. 4.5, Sec. 5.3, Appendix A.1, and the Discussion (*Buffer-ratio sensitivity*).
>
> Augmented-replay mechanism, not the specific ratio, drives the gains. No variant dominates every ordering. On **Atari** (no shared structure) the sweep traces a stability–plasticity axis: shifting capacity from FIFO toward LTDM reduces forgetting at the cost of forward transfer. On **CoinRun** (shared structure) forgetting is already low across all variants, so the spread between ratios is small and no single ratio dominates, different ratios lead on different metrics.
>
> We retain AR50: competitive everywhere without being the strongest extreme on either axis, a robust default, not a tuned optimum. Workload-aware allocation is a natural extension, now in *Buffer-ratio sensitivity*.
>
> ### 3: Reward scaling, when does it break down and why?
>
> The methods now explain the mechanism: advantage magnitudes scale with reward magnitudes, so gradient updates are dominated by the highest-return tasks. We cite PopArt-style return normalisation and DreamerV3's symlog reward standardisation, and note that per-task reward magnitudes in our Atari suite span three orders of magnitude, large enough that learning on lower-magnitude tasks suffered without per-task scaling. Static, linear, per-task scales from single-task baselines are justified by: (a) interpretability, (b) no task IDs at training, (c) non-linear advantage rescaling hurt single-task performance. Specific to suites without shared structure; CoinRun variants share a reward structure and need no scaling. The limitations flags the trade-off, with PopArt-style normalisation a natural drop-in when offline calibration is impractical.
>
> ### 4: Sample efficiency on shared-structure tasks
>
> We note the reviewer's observation about sample efficiency and expand on this. Both ARROW and DreamerV3 use replay; the difference is *what* is replayed. DreamerV3's buffer is recency-biased; ARROW mixes prior-task transitions into every update, the same mechanism that produces retention gains. Re-reading the CoinRun curves, the picture is more nuanced than originally stated. In the default order ARROW and DreamerV3 look broadly similar: both fluctuate, peak around the same level, and drop, though DreamerV3 crosses the 85% threshold first. In the reversed and two-cycle orders ARROW is more consistent, reaches its plateau sooner, and rises higher overall. We updated Sec. 5.2.1.
>
> ### 5: Anonymous code link for reproducibility
>
> The anonymous code repository (https://anonymous.4open.science/r/ARROW-B6F2/) was in the submitted manuscript but referenced only via a footnote at the start of Sec. 3, easy to miss. We moved the link to the end of the abstract and the end of Sec. 1.4 ("Our contribution").

---

### Review · Reviewer_2k8v · 2026-03-17

**Summary Of Contributions:**

The paper considers the setting of model-based continual RL, in which a world model is trained on off-policy trajectories and a policy is updated using imagined trajectories from the world model. Methodologically, the approach builds off of DreamerV3, but addresses a fundamental flaw: in DreamerV3, the replay mechanism used to train the world model is a FIFO buffer, which is biased towards recent experiences. While this is fine in the case of a single RL task — and likely even desirable given that the policy tends to improve over time — it is ill-suited to continual RL. In essence, a recency-biased replay buffer will result in hyper-specialization to the current task, forgetting past ones and harming both backward and forward transfer to other tasks. The key innovation in the proposed method, ARROW, is to dedicate a part of the replay buffer to older experiences from the agent’s entire history. As a result, the world model faces pressure to retain knowledge necessary to unroll the dynamics of past tasks, and becomes more of a generalist.

Experimentally, the authors investigate two meta-task settings: one in which tasks share structure, and one in which they do not. In both cases, they found that ARROW reliably improves task retention, backward transfer, and forward transfer (at the expense of slight drops in peak performance for the current task, at times).

My overall impression of the work was quite positive. Albeit a rather minor methodological modification to standard DreamerV3, the authors convincingly showed that DreamerV3 is ill-suited to the continual RL setting and that the long-term replay buffer alleviates much of the problems. As a whole, the paper was well-motivated, well-written, and rigorous on the experimental side.

**Audience:**

Yes

**Audience Explanation:**

- The idea is natural, effective, and unexplored. In particular, efficiently learning a world model online under reasonable memory constraints seems like a natural problem that RL must solve in order to continually learn, and the proposed method seems like a natural way to go about solving this problem.
- Ultimately, the fundamental novelty on the methodological side (from what I can gather) is just the addition of a long-term replay buffer to DreamerV3, which covers the entire history uniformly. While the results might be better and the long-term buffer might be essential for continual RL, this is a relatively small modification, which in my view limits the novelty to some extent.
- On the other hand, from the experimental side, it is quite useful to see how the long-term buffer is important for continual RL. This I think has to potential to make the paper impactful, even if the methodological novelties were rather minor.

**Broader Impact Concerns:**

N/A, I don’t think this is a concern for this paper.

**Claims And Evidence:**

Yes

**Claims Explanation:**

- The connection to neuroscience is, in my opinion, made at the right level of abstraction, seeing cortex as a world model and the hippocampus as a replay mechanism for recent experiences. I think this lends support to the general approach, since it integrates understanding from neuroscience at the algorithmic level rather than trying to replicate low-level details.
- The experiments are systematic and the suite of evaluation metrics covering both performance and efficiency is complete and clearly-defined. Overall, this makes it easy to interpret the results.
- The baselines seem sound, and the use of two kinds of meta-environments — one with shared structure and one without — is a good way to cover both extremes of continual RL.
- The impact of the key methodological intervention — the use of a long-term replay buffer — is clearly demonstrated in the results. The evidence that a long-term replay buffer is essential for model-based continual RL (at least when working within the DreamerV3 framework) is convincing.
- I think a much more convincing line of experiments would have been to explore task sequences with natural curricula; same domain / modality, but involving increased difficulty or increasingly more skills. This is where I would expect long-term replay to be most useful, because it could reinforce a good foundation of base skills, whereas I would expect DreamerV3 to overfit a specialized set of behaviours to each current task. I don’t think the current line of shared-structure tasks quite captures this, since there’s not really a natural curriculum involved.
- The paper is very clearly written. I feel like everything was presented in a logical order, and well-explained without being unnecessarily verbose. The figures are also very polished and easy to understand. As a whole, I think that it’s easy to understand the motivation, the claims, the methodology, and the relationship between the results and the claims.

**Requested Changes:**

- **Minor improvement**. Currently, the results figures split the settings into panels A, B, and C. Given that the axes are identically labeled across these settings, one necessarily has to go to the figure caption to understand the differences across figure panels. It would be a bit easier if the panels themselves simply had titles for the settings so that the figures remain more self-contained.
- **Minor improvement**. Given that there’s not much of a curriculum between the tasks, I see the “reverse task order” setting as somewhat analogous to a random seed to make sure the results aren’t coincidentally specific to a single arbitrary task order. Since the reverse task order always entails qualitatively similar results, I would suggest removing these results and discussions around them from the main paper, and just putting them in the supplementary material with a brief pointer early on in the results section. This would help declutter the figures a bit. In addition, if you were to do this, why not just explore N randomly sampled task orders, instead of just one arbitrary one and its reverse?
- **Minor improvement**. In my opinion, the two-cycle setting is the most natural and realistic. Agents generally switch between tasks, and that’s why it’s important to remember them. I would put more emphasis on this setting, presenting its results first.
- **Minor improvement**. I would be interested to see results when you spend more/less time in each task. In general, I would expect ARROW to be rather stable across these variations, but for DreamerV3 to suffer at the extremes: too much time in each task would translate to more hyper-specialization and worse forgetting/transfer, and too little time in each task would result in worse performance because the agent has too little time to learn the current task and does not get solid transfer from past ones (or from past instances of seeing that same task, in the N-cycle case).

---

> ### Author Response · Authors · 2026-05-15
> **Response to reviewer 2k8v**
>
> ### 1: Add panel titles to result figures
>
> We understand the request and did generate multiple versions. On balance, we thought that the current format was the clearest way to present the figures and note that it is a common approach.
>
> ### 2: Reverse task order, move to supplementary or sample more orderings?
>
> We retained the reversed-order experiments in the main body for two reasons. First, prior work (Rahimi-Kalahroudi et al., 2023, Appendix G) shows task ordering can produce qualitatively different forgetting profiles; in our results the reversed order substantially shifts ARROW's forgetting on both Atari and CoinRun. Second, in a CL paper the reader is best served by seeing order-sensitivity directly.
>
> On sweeping N random orderings: default + reverse is deliberately chosen to bracket the space rather than sample within it. The qualitative robustness of our findings is evidence that the conclusions are not artefacts of one arbitrary curriculum. A larger sweep would be more rigorous still, but is one of several worthwhile extensions; within available compute, default + reverse is appropriate for this submission. We make the "two opposite orderings as coverage" reasoning explicit in Sec. 4.2 and note random-ordering sweeps in Future Work.
>
> To address figure clutter (R2 and R3), we improved the textual discussion of the reversed-order results in Sec. 5: we describe the reverse-order plots primarily as confirmation that the default-order conclusions are not order-specific, reserving detailed numerical commentary for the main settings.
>
> ### 3: Two-cycle is the most natural setting; emphasise it more
>
> We agree the two-cycle setting is closest to many real deployments, and is a setting where ARROW's behaviour is distinctive (Max-F on both Atari and CoinRun is essentially zero, while DreamerV3 is substantially worse on both). We have kept the presentation order (default, reversed, two-cycle) for consistency with the bulk of the CL literature, which doesn't report on multiple cycles. To put appropriate emphasis on this setting, we added a dedicated paragraph to the Discussion highlighting its practical significance and the deployment scenario it corresponds to. The Conclusion now also calls out two-cycle behaviour as a practically relevant outcome.
>
> ### 4: Sensitivity to time spent per task
>
> We agree with the reviewer's prediction (ARROW relatively stable; DreamerV3 likely worse at both extremes). Even a minimal sensitivity sweep, pinning suite and ordering and varying only per-task budget, requires a full training run per model per seed, i.e. significant compute. We have added this to Future Work, framed as the reviewer suggests so the prediction is on the record.
>
> ### 5: Task sequences with natural curricula
>
> We agree that a domain-matched curriculum (same modality, increasing difficulty or stacked skills) would be a valuable setting. Our shared-structure setting (CoinRun visual variants) probes transfer across perturbations of a shared base environment but does not constitute a graded difficulty curriculum in which simpler skills compose into more complex ones. Together they bracket the regimes but neither *is* a natural difficulty curriculum. We have added a paragraph to the Discussion that draws this distinction explicitly, and added curriculum-style task sequences (e.g. graded Procgen difficulty levels) to Future Work. We share the reviewer's prediction: because ARROW's long-term buffer is task-ID-free and distribution-matching, prior skills can act as compositional foundations for later tasks rather than merely being retained, making a true curriculum a setting where we expect ARROW to do well.
>
> ### 6: Methodological novelty is limited
>
> We agree that the methodological intervention, augmenting DreamerV3 with a long-term, distribution-matching replay buffer, is simple. We view this as a feature: Reviewer 3 noted it positively ("the augmentation is simple (good thing!)") and we share the reviewer's own view that the empirical contribution is what matters here. The novelty we claim is not in the sampling primitives (reservoir sampling and rollout splicing are not new) but in *demonstrating that operationalising the CLS-theory split (fast hippocampal short-term + slow cortical long-term) inside a state-of-the-art world-model agent is, by itself, sufficient to convert a strongly recency-biased system into a competent continual learner under a fixed memory budget*. We have edited the Contributions 1.4 and Sec. 3.3 to make this framing explicit: the contribution is a working recipe and the evidence that the recipe is what produces the gains, rather than a new algorithmic primitive.

---

### Review · Reviewer_dwyC · 2026-04-04

**Summary Of Contributions:**

The paper presents an approach to augment the existing Dreamerv3 algorithm with an alternative replay mechanism that balances short-term recent experience and long-term knowledge in a memory-efficiency manner designed for continual learning. The augmentation is simple (good thing!), the paper considers a wide variety of metrics related to continual learning in addition to general RL performance, and evaluates the algorithm against Dreamerv3 and SAC on two sets of benchmarks (Atari and Procgen for continual learning / transfer learning.

**Additional Comments:**

It is good that the paper does not claim strong RL performance because this does seem to be lacking, especially in Figure 5. In Figure 5, it is not clear that ARROW is actually better in performance than Dreamer, even though ARROW may have more favorable stability-plasticity metrics. This is also surprising since I would imagine that shared structure would mean that better use of data could lead to better performance.

An issue seems to be that the paper is highly focused on stability-plasticity metrics, which is great to include, but arguably, we care about RL performance at the end. The diagnostic metrics should be used to explain the better performance, but it's a bit odd to have good diagnostic curves but still OK performance. Personally, I am not sure how much to weigh this since the paper does not claim to achieve strong RL performance (which is fine), but also it seems like we would always want that at a minimum. There are still potentially useful things to gain from the paper, but these are my thoughts on this point.

**Audience:**

Yes

**Audience Explanation:**

Yes, I believe so. The algorithm is simple enough that I believe people could improve upon it, and the inclusion of various stability-plasticity metrics along with the evaluation protocol is useful to build upon for other researchers.

**Claims And Evidence:**

Yes

**Claims Explanation:**

Yes, claims are supported. Generally, I would say the claims are:
1. Strong performance on a variety of continual learning metrics (such as forgetting, stability etc).
2. Good performance using a smaller replay buffer and using the data strategically compared to the naive FIFO-only buffer.
3. Evaluating algorithms on different continual learning setups.

Evidence:
1. Regarding 1, Figure 4 and 6 (and the corresponding discussions) address this.
2. Regarding 2, this is implicitly addressed in all the plots (**While the above is met, I do have comments that I have included in the Request Changes section.**)
3. Regarding 3, this is addressed in the empirical section.

**Requested Changes:**

The following change requests influence my recommendation for acceptance:
1. While it is great that the algorithm is compared against Dreamerv3 and SAC, which do not do any form of efficient data sampling, it is a bit surprising that there are no other competitive baselines that do smarter sampling for continual learning. The paper lists some related work in Section 1.2 and also mentions that ARROW can be integrated with these existing works (in Section 6), so some justification is needed as to why other approaches to deal with continual learning are not suitable baselines here. Even alternative sampling strategies like prioritized experience replay should be discussed (again, even if not directly relevant).
2. A claim in the paper is that the approach is more memory efficient. I think the authors need to define what this means. I think the claim is implicitly satisfied (i.e., smaller replay buffer, 50% of it used smartly with reservoir sampling and splicing), but if this is such a critical component, are there ways to stress test this? For example, does efficiency remain for varying sizes of the buffer (instead of fixed total $2^{19}$)? Are there alternative sampling strategies that would hurt performance? The paper does not give a clear explanation or understanding of why this specific design results in the metrics we see and which parts of the design are critical.
3. Admittedly, I am unfamiliar with the literature in Section 1.2. However, I think more justification is needed to explain why these prior works are inadequate for what the paper wants to achieve. Right now, it just lists what prior work does, but if they are, as stated, state-of-the-art approaches for continual learning, it needs to be specified why they are inadequate (related to point 1 above).


Other changes to make paper stronger:
1. It would be beneficial to state that Section 3.1 and 3.2 are taking an existing idea and are not part of the contributions. The main contribution is Section 3.3, as per my understanding.
2. Due to so many graphs, it would be helpful to explicitly point to specific plots/numbers that substantiate the claims. The paper already does this well, but there are some missing places like the second paragraph in Section 5.1. It would be helpful to re-visit wherever claims are made and include relevant references to the plots.

---

> ### Author Response · Authors · 2026-05-15
> **Response to reviewer dwyC**
>
> ### 1: Why no comparison to other CL methods (CLEAR, PER, etc.)?
>
> Rationale was underspecified; revised Sec. 1.2 and Sec. 4.2 now make it explicit.
>
> The CL families in Sec. 1.2 target a different axis than ARROW. Parameter-regularisation (EWC) and modular (PathNet, P&C) methods act on policy/critic parameters rather than the replay distribution, and require task IDs or boundaries that ARROW does not assume; they are complementary rather than alternative baselines, and can be layered on top. The family on the same axis as ARROW (buffer-side replay distribution) is replay methods, of which **CLEAR** is one example.
>
> **CLEAR** (Rolnick et al., 2019) could be compared; the obstacle is scale. Its published evaluation uses replay capacities and training-frame counts roughly an order of magnitude above ours, with the authors themselves attributing degradation at the smallest setting to over-fitting on limited stored examples. Running CLEAR at our budget would extrapolate it outside its tested range.
>
> **Prioritised experience replay (PER)** is a sampling rule, not a CL method; it slots into the FIFO half of our buffer. Revised Sec. 1.2 separates *what to keep* (reservoir + splicing, our contribution) from *how to sample* (PER and variants), and flags PER on the FIFO half as a natural follow-up.
>
> Related work now covers post-2021 model-based CL (Kessler et al., 2023; Rahimi-Kalahroudi et al., 2023).
>
> We believe that DreamerV3 and TES-SAC remain the most appropriate same-budget baselines: DreamerV3 isolates the buffer contribution from the model architecture; TES-SAC is the strongest model-free agent that fits our budget. The rationale is now explicit in Sec. 4.2.
>
> ### 2: Memory-efficiency claim should be stress-tested
>
> The new buffer-ratio ablation (R1.2) probes which design parts are critical. Sec. 4.3 now clarifies "memory-efficient": ARROW operates at half the per-buffer memory used by DreamerV3 in the original DreamerV3 setup, and is matched against DreamerV3 at our experimental budget — *like-for-like at fixed total budget*. The reviewer also asks about varying the **total** buffer size, a separate question from the FIFO/LTDM ratio. We agree this is a valuable stress test, added to Future Work; the ratio ablation is the closest evidence in this submission.
>
> ### 3: Justify why prior CL methods in Sec. 1.2 are inadequate
>
> Addressed jointly with R3.1: Sec. 1.2 now closes with an explicit paragraph stating, for each family (parameter regularisation, modular, model-free replay), what assumption or mechanism makes it unsuitable as a same-budget, task-ID-free, model-based alternative, hence DreamerV3 + TES-SAC as our control set.
>
> ### 4: Be explicit that Sec. 3.1 and 3.2 are not contributions
>
> Sec. 3 now opens stating that Sec. 3.1 (World model) and Sec. 3.2 (Actor-critic controller) recapitulate DreamerV3, and that **the contribution begins at Sec. 3.3** (augmented replay buffer), with Sec. 3.4 (task-agnostic exploration, reward scaling) as supporting design choices.
>
> ### 5: Tighten claim → evidence pointers in the results
>
> Sec. 5 now adds explicit figure-panel references wherever a quantitative claim is made, with an umbrella pointer to the detailed table at the top of each subsection. Sec. 5.1 opens with a pointer to the relevant appendix table of detailed results and then cites the relevant figure panel for each claim; the same pattern is used in Sec. 5.2 and 5.3.
>
> ### 6: Stability-plasticity metrics look good but raw RL performance is only OK
>
> We appreciate this comment and have addressed it in two places. First, Evaluation metrics (Sec. 4.4) now states up front that performance in continual RL is multidimensional and is evaluated along three complementary axes: per-task return curves ground the comparison in absolute task performance, end-of-training outcomes (ACC, min-ACC, WC-ACC) capture the deployment view, and forgetting/forward transfer characterise dynamics, with recovery and Max-F added for the two-cycle setting. Each axis answers a different question, so we read methods across the suite rather than rank on a single number, and flag where metrics mislead; most notably, low forgetting under low absolute return.
>
> Second, we agree that on individual task curves ARROW does not always beat DreamerV3 in raw return. The revised Discussion states explicitly that ARROW's trade-off, slightly lower peak per-task return for substantially better cross-task retention, is appropriate for many deployment scenarios.

---

### Comment · Action_Editor_ZGWM · 2026-03-06

Dear Authors,

Please do not upload any revisions until you have interacted with the reviewers.

Yours sincerely,

AE

---

### Author Response · Authors · 2026-04-01
**Number of reviews**

Hi, we were waiting for 3 reviewers to comment. Can we confirm that there is another review coming?
Thanks

---

> ### Comment · Action_Editor_ZGWM · 2026-04-02
>
> Yes

---

### Author Response · Authors · 2026-04-16
**Response extension request**

Thanks for reviews. They provide thoughtful and helpful feedback.
As a result of the feedback, we are conducting additional experiments to improve the paper and include in our response.
These types of experiments are very time consuming. Therefore, can we please have an extension in responding to the reviews?
Thanks

---

> ### Comment · Action_Editor_ZGWM · 2026-05-10
>
> Dear Authors,
>
> When will you upload the final revision to your manuscript?
>
> Thanks,
>
> AE

---

> > ### Author Response · Authors · 2026-05-11
> > **Manuscript revision**
> >
> > Hi, thanks for your patience.
> > We will submit it by the end of this week.
> > The experiments are finished and we updating the paper and responses.
> > Regards

---

### Author Response · Authors · 2026-05-15
**Response and revision**

We thank all three reviewers for their careful and constructive reading. The reviews converged on several points (buffer ablation, justification for baseline choice, related-work coverage, presentation), and we have tried to address each substantively. The most substantial changes are:

1. A new **buffer-ratio ablation** (25/75 and 75/25 splits between the FIFO and LTDM buffers) on Atari and CoinRun under all task orderings. The setup is described in Sec. 4.5 and results in Sec. 5.3 and Appendix A.1; the takeaway is folded into the Discussion under Buffer-ratio sensitivity. This directly addresses Reviewer 1's request for buffer-ratio ablations and Reviewer 3's question on whether the design choice is critical.
2. An **expanded discussion of baseline choice**: why TES-SAC and DreamerV3 are appropriate same-budget baselines, why CLEAR is less suitable for our specific claim (it tests a different hypothesis, and our $2^{19}$-observation budget is roughly an order of magnitude below CLEAR's smallest validated regime), and why prioritised-replay methods are orthogonal rather than competing.
3. A **rewritten related-work paragraph** that more clearly states why prior parameter-regularisation, modularity, and model-free replay methods are not direct drop-in alternatives in our setting.
4. **Presentation improvements**: labels on plots, explicit pointers from claims to plots/tables, and a clearer demarcation between background and our contribution.

**A note on updated numbers.** While preparing this revision we identified an error in the reward-normalisation pipeline used to produce the submitted results: returns were being normalised against a pool that included single-task baselines from *all* methods, rather than against the ARROW single-task baseline alone, as intended. We have corrected this and re-generated every table and figure in the manuscript. The qualitative picture is unchanged: ARROW remains best on the stability--plasticity headline metrics (WC-ACC and min-ACC) in every one of the six settings, and the relative ordering between ARROW and DreamerV3 on Forgetting is preserved throughout. A small number of secondary-metric rankings shift. Most notably, on CoinRun two-cycle, TES-SAC now edges out ARROW on Max-F (−0.010 vs.\ 0.023, both essentially zero) and on per-cycle forward transfer. Absolute values across all tables have shifted slightly and the performance curves are now vertically compressed. All numerical references are to the corrected results.

We also respond in detail to each individual review.

**A revised copy is uploaded, with changes in blue.**

---

### Decision · Action_Editor_ZGWM · 2026-05-17

**Recommendation:** Accept as is

**Additional Comments:**

The accessible source code and thorough rebuttal strongly warrant a Reproducibility Certification.

**Audience:**

Yes

**Audience Explanation:**

See box above.

**Claims And Evidence:**

Yes

**Claims Explanation:**

The reviewers and I agree that the paper addresses an important scalability challenge in model-based continual reinforcement learning and presents a simple, practical, and potentially broadly useful approach to replay-buffer design.

The updated manuscript is substantially stronger after the authors’ revisions. The reviewers and I particularly appreciate that the rebuttal and changes addressed prior concerns, including the new ablation study on buffer ratios, the clarification of reward scaling, and the improved contextualization of baseline failures under strict memory constraints. These additions make the empirical analysis more rigorous and clarify the stability-plasticity trade offs between the short term FIFO and long term LTDM buffers.

Overall, the reviewers and I find the contribution pragmatic and valuable to the continual learning community. They highlight the method’s memory efficiency, the strengthened evaluation under constrained settings, the use of appropriate retention and performance metrics, and the availability of source code. The submission is therefore recommended for acceptance.